# A wireless battery-free eye modulation patch for high myopia therapy

Tianyan Zhong [1], Hangjin Yi[2,3], Jiacheng Gou[4], Jie Li[3], Miao Liu[3], Xing Gao[3], Sizhu Chen[3], Hongye Guan[1], Shan Liang[1], Qianxiong He[2], Rui Lin[1], Zhihe Long [5], Yue Wang[4], Chuang Shi [4], Yang Zhan [6], Yan Zhang [1], Lili Xing [1], Jie Zhong [2,3] ✉ & Xinyu Xue [1] ✉

The proper axial length of the eye is crucial for achieving emmetropia. In this study, we present a wireless battery-free eye modulation patch designed to correct high myopia and prevent relapse. The patch consists of piezoelectric transducers, an electrochemical micro-actuator, a drug microneedle array, μ-LEDs, a flexible circuit, and biocompatible encapsulation. The system can be wirelessly powered and controlled using external ultrasound. The electrochemical micro-actuator plays a key role in precisely shortening the axial length by driving the posterior sclera inward. This ensures accurate scene imaging on the retina for myopia eye. The drug microneedle array delivers riboflavin to the posterior sclera, and μ-LEDs' blue light induces collagen cross-linking, reinforcing sclera strength. In vivo experiments demonstrate that the patch successfully reduces the rabbit eye's axial length by ~1217 μm and increases sclera strength by 387%. The system operates effectively within the body without the need for batteries. Here, we show that the patch offers a promising avenue for clinically treating high myopia.

The World Health Organization predicts that by 2050, half of the world's population will suffer from myopia. Of these, high myopia cases, with vision worse than −5.0 D, will make up about 20%, or 911 million individuals[1]. A significant number of these myopia cases involve progressive high myopia, for which standard optical corrections like glasses, orthokeratology, LASIK, ICL, IOL, and corneal refractive surgery do not effectively manage. While these traditional treatments correct refractive errors, progressive high myopia frequently deteriorates further despite these interventions, leading many patients to develop pathological symptoms[2–7]. These pathological symptoms often encompass severe ocular deformation and alterations in the retina, choroid, and sclera, resulting in visual field defects[8–13]. As pathological myopia progresses, it significantly heightens the risk of blindness for these patients[14–17]. Two primary methods are commonly used to treat progressive high myopia. The first is posterior scleral cross-linking, a chemical method using drugs and light to strengthen the sclera and control eye axis growth. This technique has shown promising results in animal studies, effectively enhancing the strength

of the posterior sclera[18]. The second method is posterior scleral reinforcement (PSR), which uses various materials to reinforce the posterior sclera. PSR has been used for over 20 years to treat high myopia and related macular conditions and is recognized as a safe and effective approach[19]. Studies suggest that PSR can relieve conditions such as myopic macular splitting and help reattach the macula, reducing the need for further intraocular surgeries[20,21]. It offers lasting support to prevent additional growth of the eye axis, making it a crucial treatment for high myopia patients with initial fundus abnormalities and more advanced stages[22,23].

Progressive high myopia, marked by continuous axial elongation of the vitreous cavity and thinning posterior collagen tissue, requires treatment focused on these changes[24]. Current surgical methods, however, face challenges. In posterior scleral cross-linking, fully exposing the posterior pole sclera during surgery proves difficult, often limiting the procedure to the equatorial region and affecting its efficacy[25,26]. Alternative methods using optical fibers for collagen crosslinking in the posterior pole encounter inefficiencies due to poor

adhesion and blood flow obstructions[27]. PSR demands extensive intraoperative exposure, increasing surgery complexity and duration. Moreover, PSR's effectiveness depends on the surgeon's skill, lacking precise intraoperative measurement. To address these challenges, a multifunctional therapeutic device needs to be designed. It's compact, easy to implant, and omits bulky, toxic batteries. The device utilizes a wireless power supply, facilitating complex treatment steps externally and simplifying surgery. It features a flexible, innovative micro-pump to adjust the eye's axial length (AXL). This pump is efficient and compatible with flexible materials, making it suitable for this purpose[28–30]. The device also includes a drug delivery system for posterior scleral reinforcement, reducing surgical complexity and enhancing drug delivery efficiency[31–33].

Here, a multifunctional therapeutic patch has been developed to address the limitations of traditional high myopia treatments. This patch combines the benefits of scleral cross-linking and PSR, providing a wirelessly controlled, battery-free solution. It includes piezoelectric transducers, an electrochemical micro-actuator, a drug microneedle array, μ-LEDs, a flexible circuit, and biocompatible encapsulation, all integrated into a compact, wireless design. Positioned on the sclera near the optic nerve, corresponding to the macular area, the patch uses piezoelectric transduction to convert external ultrasound into electrical energy. This energy powers and controls the device's components. The micro-actuator creates gas bubbles, causing a membrane to expand and retract the sclera behind the macula, effectively shortening AXL for vision correction. The microneedle array delivers riboflavin to the posterior sclera, while μ-LEDs induce scleral collagen cross-linking (SCXL), strengthening the sclera against high myopia-induced relaxation. In vivo rabbit experiments demonstrate the patch's clinical potential, with results showing significant scleral reinforcement. This technology offers both preventative and therapeutic benefits, particularly in managing progressive and pathological myopia by targeting axial elongation and sclera relaxation. It provides a proactive strategy to reduce the risk of severe ocular pathologies associated with high myopia. As shown in Supplementary Fig. 1, for patients with progressive high myopia without fundus lesions, the patch can be used for scleral collagen cross-linking and then removed, enhancing the posterior sclera's strength and preventing further eye axis elongation and pathological changes. In cases of established pathological myopia, the patch remains in place post-treatment, offering long-term support to the posterior eye and acting as a form of posterior scleral reinforcement to halt further deterioration. This multifunctional patch introduces an effective approach for treating and preventing high myopia, potentially diminishing the risk of severe ocular complications associated with axial elongation.

## Results
### Design and structure
Figure 1 shows the design and structure of the wireless battery-free eye modulation patch for high myopia therapy. The installation position of the patch is depicted in Fig. 1a, which highlights the intersection points of the meridiani and the equator of the eyeball, referred to as the anterior pole of the eyeball and the posterior pole of the eyeball, respectively. This study primarily focuses on optimizing the distance between two poles, namely the eye's axial length. To fix the patch on the posterior sclera parallel to the eyeball equator, a concavity is included in the middle to prevent contact with the optic nerve. The patch is positioned adjacent to the optic nerve, corresponding to the macula, and secured to the sclera using three leg tips that are sewn in place.

The ultrasound-based system facilitates the wireless conversion of electrical energy for the treatment of high myopia. The patch, designed with the utmost precision, integrates three highly sensitive receiving lead zirconate titanate (PZT) piezoelectric transducers: PZT 1, PZT 2, and PZT 3. The PZT 1 system serves as a positioning aid during

implantation surgery, while the PZT 2 system enables the adjustment of ocular axis length, which is an essential factor in addressing high myopia. It employs an ingenious configuration consisting of an inter-digitated electrode and an ionic solution, allowing for the controlled modification of the eye's AXL. By inducing electrolysis within the sealed reservoir, gas bubbles are generated, causing the attached thin membrane to expand. This expansion effectively shortens the eye's AXL, facilitating proper image formation on the retina. The interplay of these components is shown in Fig. 1b. Individuals with high myopia often exhibit weakened scleral strength, necessitating additional intervention. The PZT 3 system plays a crucial role in reinforcing the scleral tissue through photo-induced SCXL. A drug microneedle array, swiftly and uniformly delivers riboflavin into the scleral tissue. Concurrently, three blue μ-LEDs induce SCXL, as depicted in Fig. 1c. This process significantly enhances the biomechanical properties of the sclera, mitigating postoperative complications such as posterior staphyloma and retinal detachment. By combining the capabilities of the PZT 1, PZT 2, and PZT 3 systems, this system provides a comprehensive approach to high myopia treatment, addressing both the eye's AXL adjustment and posterior scleral reinforcement.

Figure 1d illustrates the structural composition of the eye modulation patch. It comprises piezoelectric transducers, an electrochemical micro-actuator, a drug microneedle array, μ-LEDs, a flexible circuit, and biocompatible encapsulation. Three PZT piezoelectric transducers, each with a diameter of approximately 3 mm and a thickness of 1 mm, serve as ultrasound energy converters and wireless control units. A centrally symmetrical arrangement of two red μ-LEDs (650 × 350 × 400 μm) is placed on the flexible circuit board for optical localization during surgical procedures. The electrochemical micro-actuator is made of a transparent solution reservoir (sealed with polydimethylsiloxane/polystyrene-block-polybutadiene-block-poly-styrene membrane; radius: 7.4 mm; thickness: ~440 μm) (PDMS/SBS). The flexible micro-fabricated circuit connects and manages all the units (Supplementary Fig. 2a, b), while two indicator μ-LEDs located at the tips of the legs indicate the working status of the patch. A flexible and transparent elastomer polydimethylsiloxane (PDMS) layer encapsulates the entire system. The fully-integrated system is shown in Fig. 1e, and the optical photograph shows the overall dimensions. The weight of the whole system is merely 0.41 g. The dimensions of the patch (4 mm in thickness and 19 mm between the two legs) are specifically designed for ease of implantation in the scleral region at the rear pole of a rabbit's eye. The compact and lightweight design of this integrated system (Supplementary Fig. 2c–f) enables seamless implantation, even in small animals like rabbits.

### Wireless powering
The eye modulation patch operates wirelessly, powered and controlled by an external ultrasound source that incorporates implanted localization, the eye's AXL adjustment, and photo-induced SCXL capabilities. Supplementary Fig. 3 shows the circuit board configuration of the external ultrasound source, while the PZT transducers (Supplementary Fig. 4a) efficiently convert the ultrasound into electrical energy using the piezoelectric effect. The operation of each individual circuit in this system hinges on meticulous control via an external ultrasonic source, as comprehensively depicted in Supplementary Fig. 3. Upon activation of the ultrasonic source, it propagates an ultrasonic signal. This signal, when directed towards the targeted receiver piezoelectric transducer (PZT) of a specific circuit, is received by the PZT, thereby triggering the activation of that particular circuit. This process enables the circuit to fulfill its designated role. To achieve optimal circuit impedance matching, it is crucial to consider the resonant frequency and other performance parameters of both the ultrasound source transducer and the receiving transducer in the circuitry. Supplementary Fig. 4b, c illustrate the resonant frequency and admittance circle diagram of the PZT receiver, indicating that the PZT

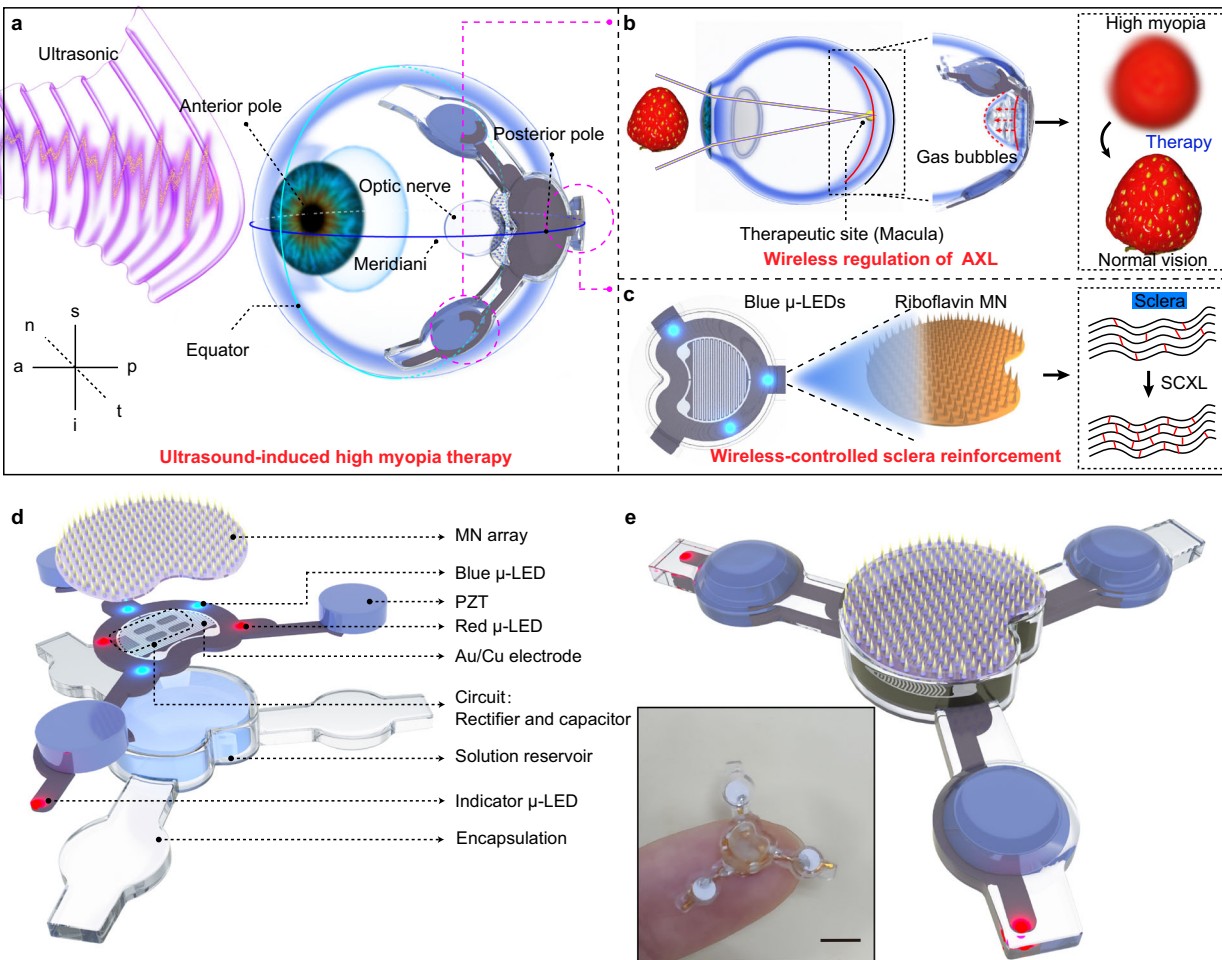

**Fig. 1 | Illustrations of the multifunction flexible eye modulation patch and subcomponents. a** Designs, operational features and use cases for wireless eye modulation patch. **b** Wireless charging based on ultrasound, adjustable micro-actuator for shortening the axial length (AXL) of the eye and improving vision to normal levels. **c** Under blue μ-LEDs irradiation, riboflavin promotes collagen cross-linking (SCXL) in scleral tissue, enhancing scleral strength. **d** Exploded view of the flexible eye modulation patch. Microneedle array (MN array). Piezoelectric transducer: lead zirconate titanate (PZT). **e** Integration of the corresponding subcomponents and optical image of the patch. Scale bar, 5 mm. a-anterior, p-posterior, n-nasal, t-temporal, s-superior, i-inferior.

transducer efficiently receives response signals during operation. Following impedance matching, the ultrasound frequency for driving the PZT transducer is 633 kHz (the corresponding electrical output is shown in Supplementary Fig. 4d). Ultrasound with this frequency has efficient transmission through the gel medium and ocular tissue. The electrical output performance is consistently stable and uniform, as shown in the inset of Supplementary Fig. 4d. The ultrasonic transducer excitation circuit is designed with an adjustable duty cycle (10–100%), as demonstrated in Supplementary Fig. 4e. In the experiment, a duty cycle of 30% is selected to minimize power loss while meeting the experimental conditions. Actually, the PZT transducer exhibits high excitation voltage at duty cycles ranging from 10% to 70%, and the duty cycle can be chosen in this range based on practical demand (Supplementary Fig. 4f). As shown in Supplementary Fig. 4g, the PZT transducer produces a relatively stable output voltage for 10 min at a 30% duty cycle. Additionally, as illustrated in Supplementary Fig. 4h, the output power of the PZT transducer remains relatively constant within a small range as the distance between the ultrasound source and the PZT transducer increases from 1 to 30 mm. Similarly, the output performance of the PZT at different angles is measured when the ultrasound source is 7 mm away, demonstrating a stable amplitude (Supplementary Fig. 4i).

Figure 2a illustrates three PZT transducers that independently power three circuits. Optical images of the three functions are shown

in Fig. 2b–d, respectively. As shown in Supplementary Fig. 5a, b, the PZT 1 circuit features two-opposing red μ-LEDs located in the center of the disc, which are used solely as optical positioning assistance during surgery and have no therapeutic effect. The inset in Supplementary Fig. 5b illustrates a side view of the system in action. The PZT 2 circuit includes a rectifier bridge and a capacitor to provide a stable DC signal for the electrolysis of ionic solution within the micro-actuator. A critical component within the bridge circuit is the indicator μ-LED. The optical image and indicator diagram of the system in operation are shown in Supplementary Fig. 6a. The PZT 3 circuit consists of three blue μ-LEDs that serve as photo-induced SCXL, and another indicator μ-LED (on the tip of the leg), indicates the working status of the system. The optical image and indicator diagram of the system in operation are shown in Supplementary Fig. 6b. The corresponding circuit diagrams for each function are illustrated in Fig. 2e–g.

The wireless powering of the patch is assessed within pork tissues at varying depths. The voltage remains consistently around 7 V, while the current ranges between 4–5 mA (Fig. 2h–i). Its stability and functionality persist within the depth range of 7–35 mm. Figure 2j depicts the current output of the location red μ-LED and cross-linking blue μ-LEDs within a distance range of 1–30 mm. Notably, even as the ultrasound source and the PZT transducer increase, there is minimal fluctuation in the circuit's current during operation. This observation highlights the system's high stability.

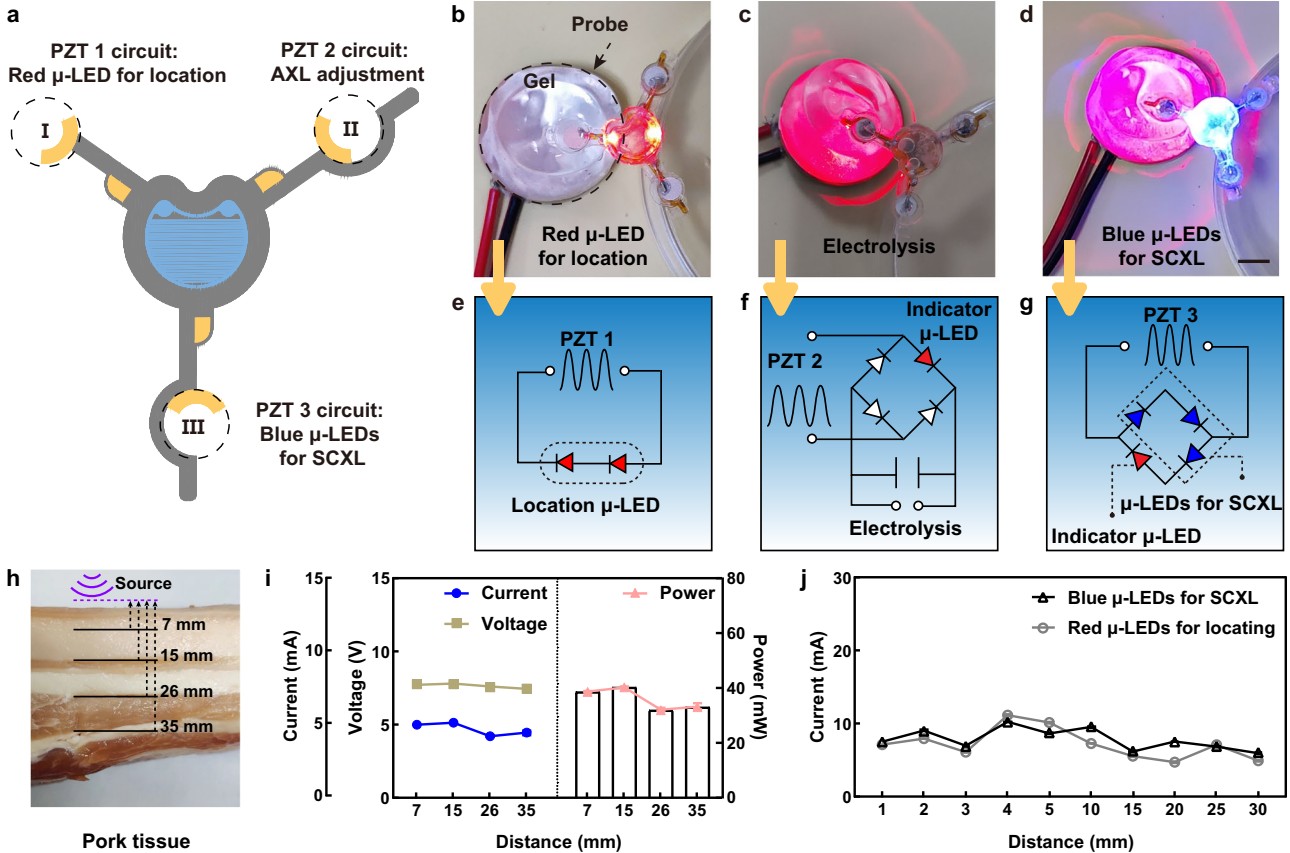

**Fig. 2 | Output performance of the receiving ultrasonic transducer. a** The planar view of a flexible electronic system. **b** The first-leg component features dual-red μ-LEDs localization, (**c**) the second-leg association with electrolytic, (**d**) and the third-leg for scleral collagen cross-linking (SCXL) under blue μ-LEDs irradiation, including circuit diagrams for (**e**) location, (**f**) electrolysis, and (**g**) cross-linking. Scale bra, 5 mm. **h**, **i** The output power at different tissue depths. Data are expressed as mean ± SD, with each experiment independently replicated 11 times, yielding consistent results ($n = 11$). **j** The working current of the location red μ-LEDs and cross-linking blue μ-LEDs at different depths in the gel. Data are expressed as mean ± SD, with each experiment independently replicated 8 times, yielding consistent results ($n = 8$). Source data are provided as a Source Data file.

## Electrochemical micro-actuator

Figure 3a schematically illustrates that the electrochemical micro-actuator receives wireless energy using ultrasonic transducers and provides force to overcome intraocular pressure, thereby facilitating the restoration of the myopia eyeball to its normal AXL. The completely wireless operation of the micro-actuator eliminates the intricacies and inaccuracies inherent in the conventional medical apparatus utilized for AXL adjustment during previous surgical interventions. Moreover, the adoption of wireless energy transmission eliminates the potential harm associated with conventional bulky batteries. The directional nature of ultrasound transmission (as shown in Fig. 3b) allows for precise control over transmission distance and energy intensity, enabling high-precision transmission. Simulation results show that acoustic pressure ($P_0$ ~ 0.6 MPa) is slightly attenuated during transmission through gel, vitreous and sclera, but remains perpendicular to the transducer surface. The effective working range is approximately a 10 mm diameter area, while the shortest distance between two adjacent transducers is larger than 15 mm (the diameter of the rabbit eyeball), efficiently avoiding the mis-triggering of other functions. Supplementary Fig. 7 shows the temporal evolution of acoustic pressure after ultrasonic transmission from the anterior pole to the posterior pole of the rabbit eye. This visualization captures the dynamic nature of acoustic pressure and the patterns and changes that occur over time. The working state of the electrochemical micro-actuator on the porcine eyeball in vitro is shown in Fig. 3c, with the indicator μ-LED indicating that electrolysis is in progress. Upon wirelessly powering the micro-actuator, the gas bubbles produced by

solution electrolysis gradually increase, as shown in Fig. 3d. The core of the electrolysis process is the use of microfabricated Cu/Au interdigitated electrodes utilized for electrolyzing aqueous solutions to produce hydrogen and oxygen. The optical image of the interdigitated electrodes is shown in Supplementary Fig. 8.

Figure 3e demonstrates that the maximum electrolysis current is achieved at an applied voltage of approximately 4.5 V. The interdigitated electrodes within the micro-actuator facilitate the electrolysis reaction, converting $2H_2O$ (liquid) to $O_2$ (gas) + $2H_2$ (gas). To ensure sufficient conductivity, a 50 mM NaOH electrolyte solution is utilized. The flexible membrane undergoes mechanical deformation due to volume expansion resulting from hydrogen and oxygen production. Supplementary Fig. 9a presents optical photographs illustrating bubble generation at different voltage, with the highest speed and density observed at 3.5 V. It is noteworthy that beyond this voltage threshold, the rate of electrolysis remains relatively constant. The patch's area is 14 mm² to cater to the specific needs of the treatment site. Considering the electrolysis duration of 6 min, a sufficient volume of electrolyte solution is essential for uninterrupted operation. In scenarios with limited electrolyte solution volume, the process may prematurely cease due to inadequate liquid to separate the bubbles from the electrodes. This insufficiency could lead to bubbles enveloping the electrode surface, thereby halting the electrolytic reaction. As demonstrated in Supplementary Fig. 9b, with an electrolyte solution volume of 15 μL, the liquid chamber's thickness measures 1.07 mm, but the reaction ceases after 270 s as bubbles cover the electrode surface. Conversely, increasing the electrolyte solution volume to

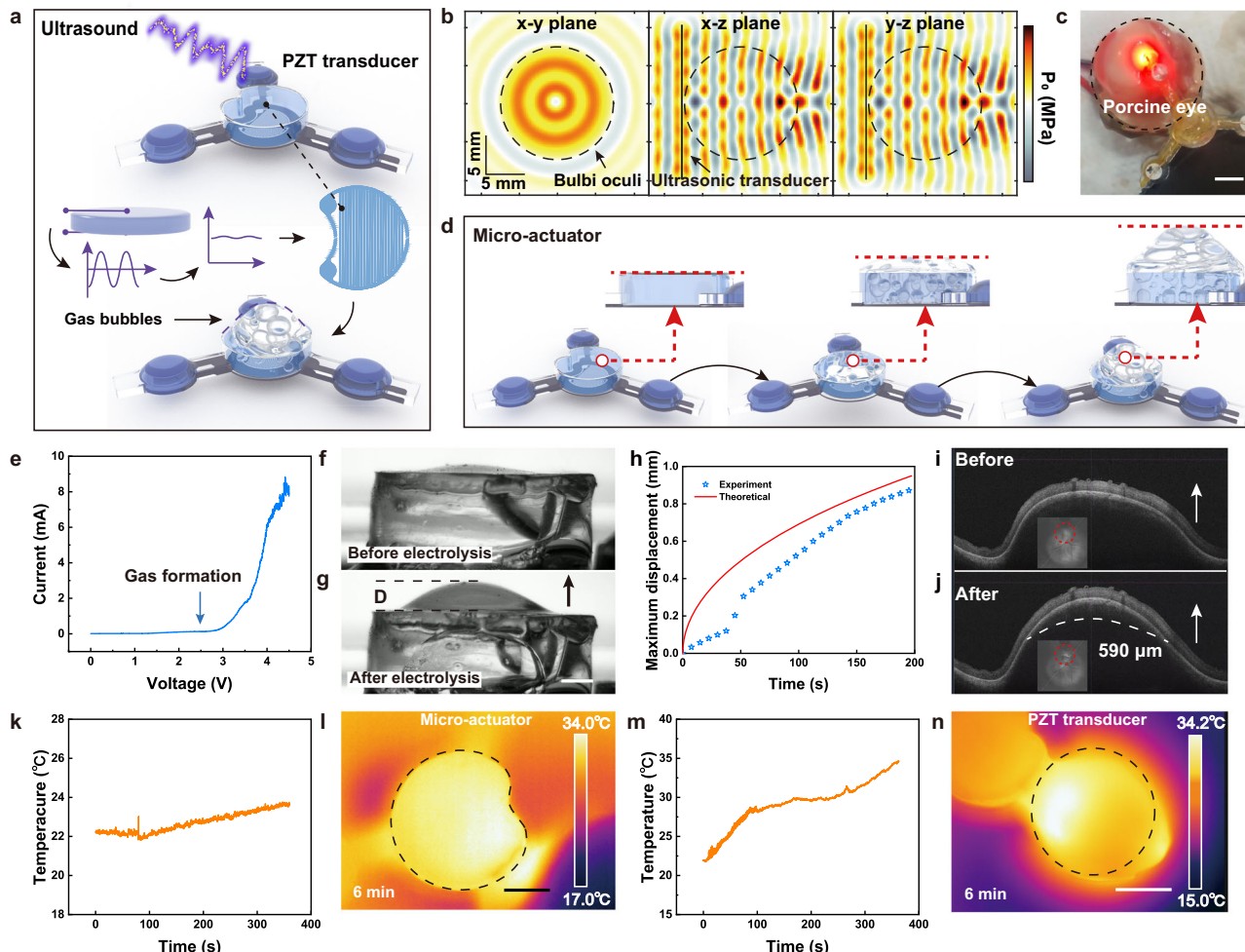

**Fig. 3 | Electrochemical micro-actuator system. a** Ultrasonic-induced electrolysis of the solution through an interdigitated electrode. **b** Simulation of acoustic pressure distribution in the eyeball. **c** Optical image of ultrasonic-induced electrolytic solution with an indicator on the porcine eyeball. Scale bar, 5 mm. **d** Functional state diagram of the micro-actuator. **e** Current-voltage characteristics of the electrochemical micro-actuator with a voltage range of 0–4.5 V. Images of mechanical deformation of the flexible membrane induced by solution electrolysis. **f** before and (**g**) after. Scale bar, 1 mm. **h** Comparison between experiment and theoretical of maximum displacement of flexible membrane induced by solution electrolysis. **i** Upon implanting the patch at the posterior pole of the porcine eye, there was a reduction in the fundamental eye axis length. **j** Subsequent electrolysis resulted in a modification of the eye axis length 590 μm. **k** Temperature variation in the micro-actuator within 6 min. **l** Infrared thermal image of micro-actuator. Scale bar, 2 mm. **m** Temperature variation in PZT transducer within 6 min. **n** Infrared thermal image of the PZT transducer. Scale bar, 2 mm. Source data are provided as a Source Data file.

25 μL results in a liquid chamber height of 1.78 mm, allowing the electrolytic reaction to proceed smoothly for up to 420 s without interruption. Our experiments indicate that the minimum volume capacity of electrolyte solution in the liquid chamber is 25 μL. To ensure the reliability and stability of the experiment, we designed the chamber to hold a volume of 35 μL, corresponding to a chamber height of 2.5 mm and an overall patch thickness of approximately 4 mm. This configuration allows for further optimization based on experimental needs to achieve optimal functionality. Figure 3f, g visually represents the expansion of the membrane in the micro-actuator. Upon activation, gas accumulation inside the sealed micro-actuator leads to increased pressure. Under typical operating conditions (applied current of ~5 mA), Fig. 3h and Supplementary Note 1 demonstrate that the maximum membrane deformation reaches approximately 870 μm within a duration of 200 s. This dynamic is further illustrated in Supplementary Movie 1. Furthermore, the reliability of the interdigitated electrode is confirmed by the cyclic voltammetry curve in Supplementary Fig. 10. In an in vitro porcine eye experiment, the eye modulation patch efficiently regulates the AXL when placed over the eyeball. The patch's implantation site is depicted in Supplementary Fig. 11, with Optos images indicating deformations in the fundus's right side, implying that the patch exerts axial pressure on the macula. Supplementary Movie 2, along with spectral domain optical coherence tomography (OCT) B-scans (Fig. 3i, j), shows the patch's ability to adjust the AXL, achieving a change of about 590 μm before and after the micro-actuator operates.

During electrolysis, the temperature fluctuation caused by the reaction is limited and can be neglected inside the body, as shown in Fig. 3k–n. The temperature of the micro-actuator increases slightly from 22.2 °C to 23.5 °C within 6 min, which is imperceptible for human beings. The temperature of the PZT transducer increases from 21.9 °C to 34.6 °C within 6 min, indicating that in practical application the ultrasonic operation time should be within several minutes. The infrared thermal imaging of the heating process of the micro-actuator and PZT transducer is presented in Supplementary Fig. 12 and Supplementary Fig. 13, respectively. The maximal temperature of the patch is lower than the body temperature, confirming the safety of the patch.

## Drug microneedle array and light-induced SCXL

Adults with high or pathological myopia often experience ongoing axial elongation of the eye, which can lead to conditions like myopic maculopathy and severe visual impairment[34]. To combat this, strengthening the posterior scleral structure after adjusting axial dimensions is crucial to prevent further elongation due to scleral laxity. Scleral collagen cross-linking, including riboflavin/UV light and riboflavin/blue light methods, is emerging as an effective strategy. This process enhances chemical bonding in collagen fibers, increasing scleral rigidity[35]. The combination of riboflavin with blue light for scleral collagen crosslinking is typically irreversible, making it apt for long-term scleral reinforcement to slow down myopia progression[36,37]. However, the dense nature of the sclera makes drug delivery challenging[38–41]. To facilitate drug absorption, meticulous exposure of the posterior scleral site is imperative within the surgical context. Continuous riboflavin infusion, along with high-intensity light, is key for initiating cross-linking. Regrettably, prolonged exposure of the posterior scleral region may lead to desiccation and attenuation of the scleral tissue[42,43].

Recent advancements in drug microneedle technology have enabled the efficient delivery of various therapeutics, including small molecules, peptides, and vaccines, through the skin[44–46]. These microneedle-based systems show great promise for delivering drugs to specific ocular areas such as the cornea[47,48], suprachoroidal space[49–52], and sclera[53–55]. Solid microneedles allow for controlled drug release at the target site. In the myopic patch, the microneedle array is placed on the targeted area, where it penetrates the sclera for rapid and even drug release, as shown in Fig. 4a. The array, made of polyvinylpyrrolidone (PVP) and riboflavin, is crafted using a PDMS mold (details in the methods section). Riboflavin's structure includes an isoalloxazine ring and a ribityl side chain (Fig. 4b), where the ring's atoms participate in redox reactions crucial for various biological functions[56,57]. Scanning electron microscopy (SEM) images (Fig. 4c, d) reveal the microneedle array's uniform arrangement.

In addition to its biological functions, riboflavin possesses unique fluorescent properties[37,58]. It can absorb light in the ultraviolet range (around 360–383 nm) and blue light range (around 420–470 nm) (Supplementary Fig. 14a, b), and emit fluorescent light in the blue/green range (with maximum peaks at 373 nm and 443 nm), making it easily detectable by fluorescence spectroscopy and microscopy (Fig. 4e, f). Figure 4e shows the structural image of riboflavin observed under a fluorescence microscope. The riboflavin molecules are rod-shaped, with a size range of 60–250 μm. Fluorescence imaging of riboflavin microneedles in Fig. 4f reveals their structure, which includes a center-to-center spacing of 600 μm, a bottom diameter of 370 μm, and a needle height of 400 μm.

Prior to the deployment of microneedle patches for in vivo transdermal bio-detection, an extensive investigation is conducted to examine their biophysicochemical properties, mechanical strength for dermal tissue penetration, and light-induced SCXL. Supplementary Fig. 15a illustrates a uniform array structure of drug microneedles following insertion into agarose. To assess the mechanical strength of riboflavin microneedles for penetrating porcine sclera under compression, a micro-compression test is performed on a microneedle patch comprising a diameter of 10 mm of microneedles. Remarkably, the microneedle patch demonstrates the ability to withstand a compression force exceeding 2.1 N per microneedle at a height of 400 μm, which is considered sufficiently high to puncture the posterior sclera of the porcine eye without encountering mechanical yielding[59]. Consequently, the drug microneedles successfully penetrated the porcine sclera, allowing direct delivery of the riboflavin drug to the desired location, thereby enhancing drug permeability and absorption (inset of Fig. 4g and Supplementary Fig. 15b). Utilizing nuclei 4',6-diamidino-2-phenylindole (DAPI) staining, the depth of microneedle penetration into the scleral tissue is clearly visualized, indicating an approximate

depth of 84 μm into the posterior sclera of porcine eyes (Supplementary Fig. 15c). Notably, no effects on the choroid and retina layers are observed. Additionally, the shallow depth of the microneedle penetration emphasizes the minimally invasive nature of the procedure. The fluorescence intensity of the posterior sclera tissues within 0–30 min after drug delivery is shown in Fig. 4h and Supplementary Fig. 16. The intensity gradually increases, indicating successful drug diffusion into the scleral tissue.

Riboflavin-UV/blue light SCXL is commonly used to strengthen the posterior sclera[58–60]. However, prolonged UV light exposure may cause eye issues like dryness, inflammation, and an increased risk of cataracts or macular degeneration[61,62]. Hence, a milder blue wavelength is preferred. The blue μ-LED, peaking at 443 nm, aligns with riboflavin's absorption peak (Supplementary Fig. 14b). The riboflavin/blue light-induced SCXL operates through a photochemical oxidation reaction[58]. When exposed to UV or blue light, riboflavin enters a highly reactive triplet state $^3RF^*$, generating reactive oxygen species such as superoxide anions ($O_2^-$) and singlet oxygen ($^1O_2$). Singlet oxygen reacts with biomolecules, including collagen fibers and proteoglycans, forming additional covalent bonds and tightening the collagen network, as depicted in Fig. 4i. Riboflavin solution also generates free radicals like hydroxyl radicals under UV/blue light, further promoting SCXL. This method effectively enhances the sclera's stability and mechanical properties.

In light of prior challenges inherent in conventional SCXL surgery, which necessitated the complete exposure of the posterior scleral region thereby heightening the susceptibility to iatrogenic injury, a solution is engineered. Integrating a blue μ-LED array into the patch and situating it at the designated therapy site, while employing external ultrasound as a modulatory instrument to precisely initiate and conclude the SCXL procedure, results in an integrated treatment system that effectively mitigates intraoperative trauma risk. Simultaneously, it enables a meticulously tuned regulation of the comprehensive cross-linking cascade. As shown in Supplementary Fig. 17a, the light intensity of the blue μ-LED is measured under different currents. As the distance between the ultrasound source and the PZT transducer is approximately 10 mm, the current passing through the blue μ-LED is 9.6 mA (Fig. 2j), and the intensity of blue light is measured to be ~32 mW cm$^{-2}$. Under this light intensity, the Young's modulus of porcine sclera after SCXL is measured to be 40.3 MPa, which increased by approximately 136.31% compared to the sclera without light exposure and drug immersion (Control and Riboflavin group) (Fig. 4j). For the riboflavin group (RF) where only riboflavin is immersed without light exposure, there is almost no change in Young's modulus compared to the control group (CTRL).

The rabbit sclera, approximately half as thick as pig sclera and containing less melanin (Supplementary Fig. 17b), shows slightly stronger light transmission compared to pig sclera (Supplementary Fig. 17c). Photographic documentation and transmittance measurements under various light intensities reveal that at around 30 mW cm$^{-2}$ light intensity prior to penetration, the transmission rate through sclera and choroidal tissues registers at 55.46% for rabbits and 15.40% for pigs (Supplementary Fig. 17d, e). The structural resemblance between pig and human eyes suggests that only low-intensity light reaches the fundus after penetration, affirming the safety of the fundus during the experiment. COMSOL simulations show that one or two μ-LEDs do not provide the necessary irradiation area and intensity for effective collagen cross-linking due to limited lighting range and location. This is evidenced in Supplementary Fig. 17f, where such configurations lead to uneven and inadequate light intensity distribution across the sclera, impeding collagen cross-linking in the target area. In contrast, arranging three μ-LEDs around the patch secures full coverage and adequate cross-linking light intensity, reaching the middle sclera layer and enhancing the cross-linking effect. The samples for Young's modulus measurement need to be stretched slightly to compare the changes

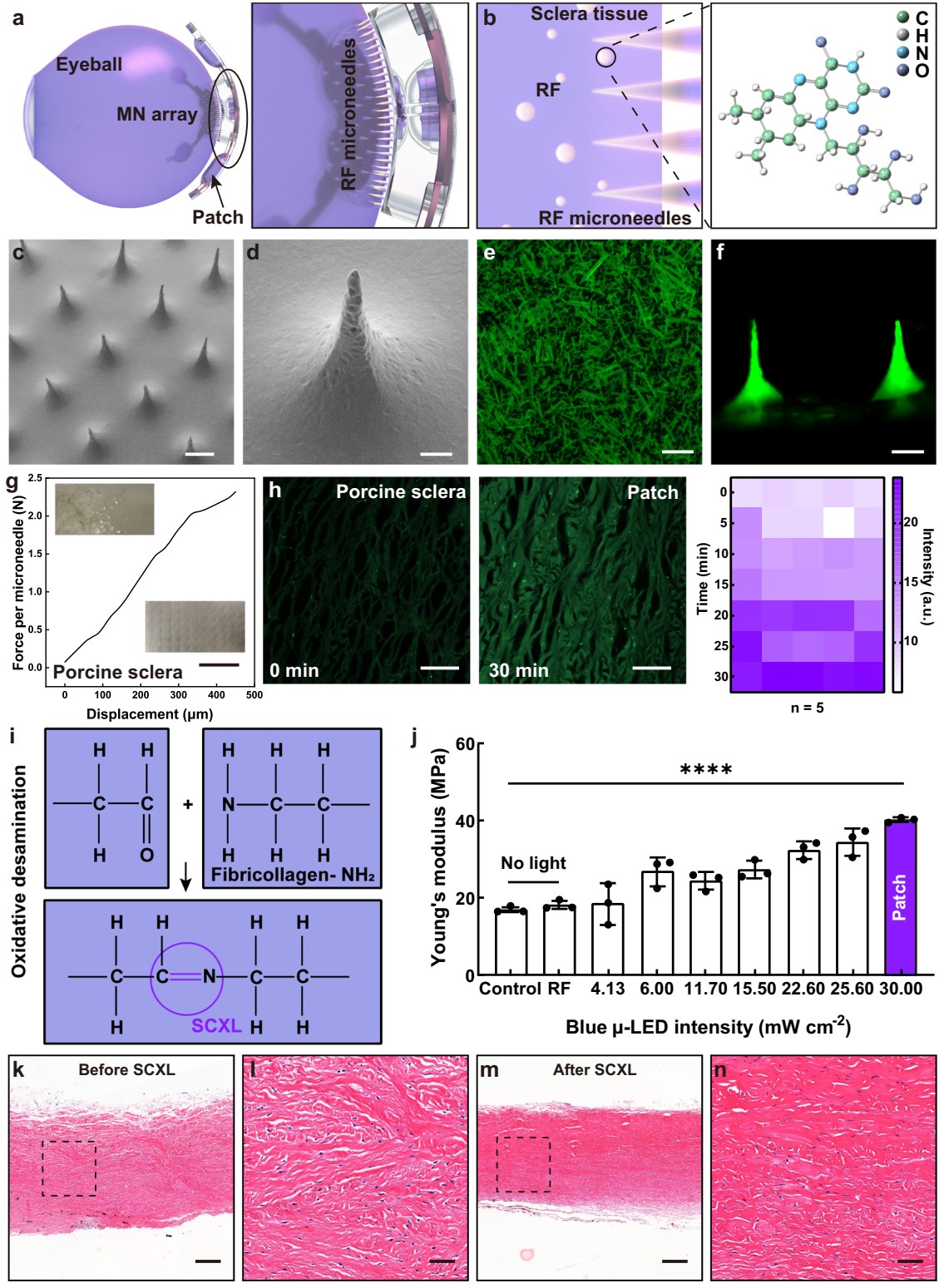

before and after treatment, as shown in Supplementary Fig. 17g. The testing range is within the linear elastic region, which is usually in the low-stress range. Hematoxylin and eosin (H&E) staining results of the porcine sclera tissue before and after SCXL are shown in Fig. 4k–n and Supplementary Fig. 18a–h. Collagen crosslinking results in a more compact and regular organizational structure of collagen fibers compared to the pre-crosslinking state. The tissue density increases, while the intercellular matrix decreases. Notably, no significant structural or degenerative alterations occur during this process.

## AXL modulation of rabbit eye in vivo

The schematic representation of the patch, as depicted in Fig. 5a, b, illustrates its utilization for eye modulation by enveloping the eyeball and regulating the AXL through the application of wireless ultrasonic transmission. Through the generation of bubbles, the internal gas pressure induces the outward expansion of the low-hardness membrane, resulting in contraction of the macular region within the posterior sclera toward the central area. As a result, the AXL undergoes a reduction, effectively achieving the desired goal of vision correction.

**Fig. 4 | Biophysicochemical properties of riboflavin (RF) microneedle array and light-induced scleral collagen cross-linking (SCXL). a** Affixing the integrated microneedle array of the myopic patch to the targeted treatment area. Right inset: Zoomed-in detail of microneedle array piercing sclera. **b** Illustration of the process of microneedle dissolving and releasing riboflavin within the sclera. Right inset: Structure of riboflavin. **c** Scanning electron microscopy (SEM) image of the microneedle array. Scale bar, 200 μm. **d** SEM image of a single microneedle. Scale bar, 50 μm. **e** Fluorescence image of the riboflavin drug. Scale bar, 200 μm. **f** Side-view fluorescence image of the drug microneedle. Scale bar, 200 μm. **g** Mechanical behavior of the riboflavin microneedle array under normal compressive load. Up inset: optical image of normal porcine sclera. Down inset: optical image of porcine sclera administered with a microneedle array, showing the indents caused by the penetration of microneedles. Scale bar, 2 mm. **h** Fluorescence images of drug release from microneedles into sclera over time (0–30 min), and changes in the

corresponding quantitative analysis of the mean fluorescence intensity (a.u.). *n* = 5 porcine sclera per group. Scale bar: 200 μm. **i** Schematic representation of the SCXL process using riboflavin and blue μ-LEDs. Riboflavin and blue μ-LEDs induce extra covalent bonds between and within collagen fibers and between collagen and proteoglycans, enhancing scleral strength. **j** The relationship between blue light intensity and Young's modulus. The control (CTRL) group represents the condition with no light and no drugs, while the riboflavin (RF) group represents the condition with no light but with drugs. Statistical comparisons were assessed by one-way analysis of variance (ANOVA), 95% confidence interval; ****$p$ < 0.0001. *n* = 3 porcine sclera per group. Data are presented as mean values ± SD. **k, l** Hematoxylin and eosin (H&E) staining of the sclera before SCXL. Scale bar in (**k**): 200 μm; scale bar in (**l**): 50 μm. **m, n** H&E staining of the sclera after SCXL. Scale bar in (**m**): 200 μm; scale bar in (**n**): 50 μm. Source data are provided as a Source Data file.

To evaluate the effectiveness of the patch implanted on the sclera of live rabbit eyes, a comprehensive assessment is performed (Supplementary Fig. 19a). The physiological parameter of right eye (OD) of each rabbit is recorded pre-treatment (control group) and post-treatment (experimental group). The patch implantation starts after opening the eyelids with a speculum (Supplementary Fig. 19b–e). We utilize a binocular indirect ophthalmoscope to observe whether the patch is accurately positioned, and turned on red μ-LEDs for localization calibration when appropriate (Supplementary Fig. 19f). X-ray images shows the location of the patch wrapped behind the right eyeball at the sclera (Fig. 5c). The X-ray image of the left eye (OS) is shown in Supplementary Fig. 19g. The optical image of rabbit eyes after patch implantation shows no obvious lesions or strabismus (Supplementary Fig. 19h). The external ultrasound is activated, initiating the operation of the patch within the fundus (Fig. 5d). The electrolysis process is indicated by the μ-LED indicator. (Fig. 5e and Supplementary Movie 3). Subsequently, the deformation in this region could be observed in the optos images (Fig. 5f), In contrast, no signs of deformation were observed in the optos images of the left eye (Supplementary Fig. 19i). The horseshoe-shaped notch part of the patch is located about 5 mm below the optic nerve to avoid squeezing it (Supplementary Fig. 19j, k).

Figure 5g–i, shows OCT B-scan images illustrating the progression: pre-treatment, patch implantation, and post-treatment, respectively. The images conspicuously display pronounced scleral bulges, demonstrating a cumulative height alteration of approximately 1030 μm post-treatment in contrast to the pre-treatment condition. OCT 3D display images (Fig. 5j) also show deformation at the posterior pole behind the sclera, corresponding to the deformation of OCT B-scans. Ocular ultrasonography (A-B Scan) vividly captures the AXL changes before and after treatment (Supplementary Fig. 20), accentuated by the high-reflection patch in the fundus oculi (indicated by arrows). Figure 5k depicts variations in AXL, revealing a mean AXL reduction of approximately 1217 μm observed across within a span of 6 min (*n* = 6). The normalization of the six data sets reveals that the adjustment fluctuation of the axial length (AXL) is relatively uniform (Fig. 5l). The AXL findings unequivocally establish that micro-actuator leads to AXL reduction, validating the patch's ability to modulate AXL. It's essential to acknowledge the potential for individual rabbit variations and the influence of surgical sutures on patch implantation adjustments. The wireless controllability of the patch allows for a secondary electrolysis session to fine-tune axial length (AXL) adjustments if initial attempts do not meet prescribed standards. The correlation between adjusted AXL and time is detailed in Supplementary Note 1,

$$EF\pi^2h^2(3a^2 + h^2)^2 = 81iRT\,\pi la^2(1 - 2\nu)t, \qquad (S1)$$

which can precise and independent control of eye axis adjustments in experimental subjects with individual variations. Eq S1 is instrumental

in representing the relationship of the AXL variation *h* and the time *t*. This capability for subsequent adjustments introduces a promising approach to high myopia treatment.

### SCXL on rabbit sclera in vivo and immunohistological analysis
After modulating the AXL of the rabbit eye, the SCXL is initiated. The system is activated using ultrasound, as depicted in Fig. 6a. During the cross-linking process, three blue μ-LEDs illuminate, and the red indicator light flashes at the same frequency. As observed in Fig. 6b and Supplementary Movie 4, purple-colored light (resulting from a mixture of red and blue light) can be seen in the rabbit eye. The cross-linking procedure lasts for 30 min. Euthanasia is performed in batches at 22 days, and measurements of Young's modulus, along with relevant pathological sections, are conducted to verify the experiment's effectiveness and safety. The measurements of Young's modulus reveal an approximate increase of 151.25% in the posterior sclera's Young's modulus 7 days after SCXL, and 387% at 22 days (Fig. 6c, d). Previous studies[18] indicate that the effects of cross-linking are enduring, showing no notable reduction in Young's modulus even after 8 months of observation. This persistence suggests that posterior scleral reinforcement is a promising approach for slowing the progression of high myopia. The control group, H&E sections (Fig. 6e) reveal a loose arrangement of collagen fibers. Conversely, in the cross-linking group, the collagen fibers display a compact alignment. Porosity analysis of H&E sections demonstrated a 58.33% reduced porosity in the SCXL group compared to the control group (Fig. 6f and Supplementary Fig. 21). This implies the effective strengthening of the rabbit posterior sclera through riboflavin-induced enhancement under blue light exposure. This reinforcement of the posterior sclera effectively contributes to the mitigation of high myopia recurrence.

In this study, the structural changes were examined in the lamina fusca (LF) and scleral stroma (Sc) before and after the SXCL intervention. Initially, in control cases (Supplementary Fig. 22a–c), the LF and Sc contain elastic fibers (e) and fibroblasts (fb), with the fibroblasts oriented parallel to the eyeball surface and characterized by elliptical nuclei (n) and thin cytoplasmic processes (p). Following the SXCL intervention (Supplementary Fig. 22d–f), the same regions display macrophage-like cells (mp) within the stroma and fibroblasts with notably thickened processes (*). Further detailed observation under higher magnification (Supplementary Fig. 22g, h) reveals thick fibroblast processes post-SXCL, rich in endoplasmatic reticulum (ER) and primary lysosomes (lp). Additionally, a frontal section of a collagen fibril bundle with collagen fibrils (cf) and elastic fibers (arrow) is compared pre- and post-SXCL (Supplementary Fig. 22i–l). Lastly, the diameter of single collagen fibrils is measured (Supplementary Fig. 22m, n), showing no significant difference between the control and post-SXCL interventions, thereby underscoring the subtle yet significant impact of the SXCL treatment on the scleral structure. A substantial amount of research indicates a close association between high myopia and glaucoma in terms of epidemiological features, clinical

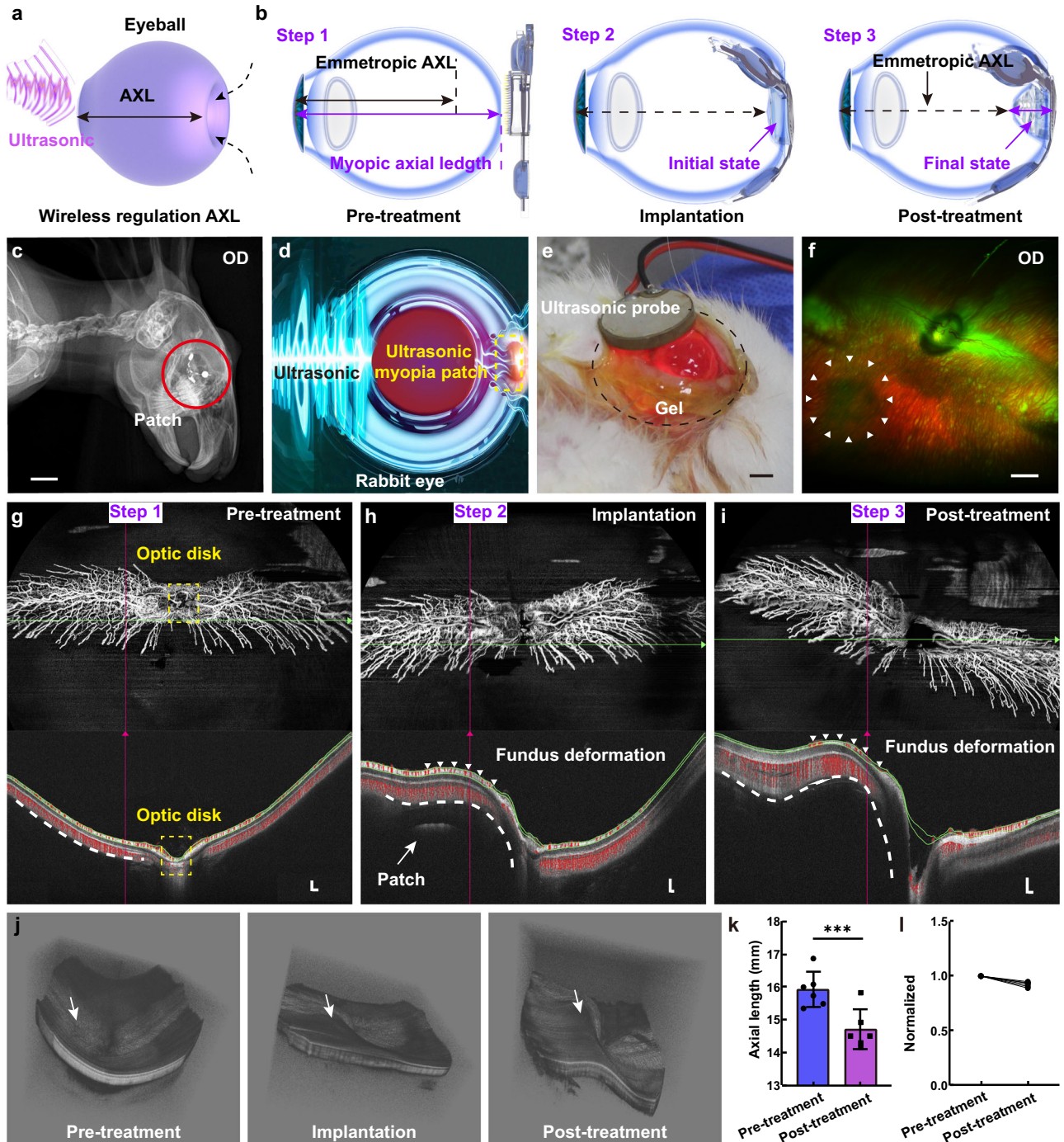

**Fig. 5 | Implantation of the wireless eye modulation patch and modulation on the axial length (AXL) of the rabbit eyeball. a**, **b** Schematic diagram of a wireless AXL adjustment system that operates on ultrasonic energy transmission. **c** X-ray image indicating the position of the patch implantation in treatment eye. Scale bar, 15 mm. **d** Ultrasonic through the rabbit eye, delivering energy to the patch positioned behind the eyeball. **e** Optical image displaying the electrolysis is initiated. Scale bar, 5 mm. **f** Deformed Optos image of the fundus generated after patch operation. Scale bar, 3 mm. **g**–**i** OCT B-scans showing images pre-treatment, after patch implantation, and post-treatment, respectively. Scale bar, 200 μm. **j** OCT 3D images demonstrating axial deformation of the fundus. **k** Comparing axial length change in the pre- and post-treatment. **l** Data are presented as mean values ± SD, using 95% confidence interval, two-sided, paired *t* test; ***p = 0.0004 (*n* = 6 eyes). Source data are provided as a Source Data file.

manifestations, and pathogenic mechanisms[63]. Therefore, maintaining normal postoperative intraocular pressure (IOP) is a crucial aspect of this study. Encouragingly, On the day of surgery, the highest recorded intraocular pressure (IOP) is approximately 23.6 mmHg, with the lowest being 16.6 mmHg. By the third postoperative day, IOP had essentially returned to its preoperative value of 10.3 ± 2 mmHg (Supplementary Fig. 23a, b). Over the subsequent 20 days, IOP fluctuated

around this normal range. This transient rise in IOP can be attributed to factors such as a diminished vitreous cavity or post-operative inflammation. Consequently, the patch appears to have an insignificant impact on the IOP of rabbit eyes during the 20-day postoperative observation period.

To assess the presence of neurodegenerative changes, we examine the morphological alterations of microglial cells and astrocytes.

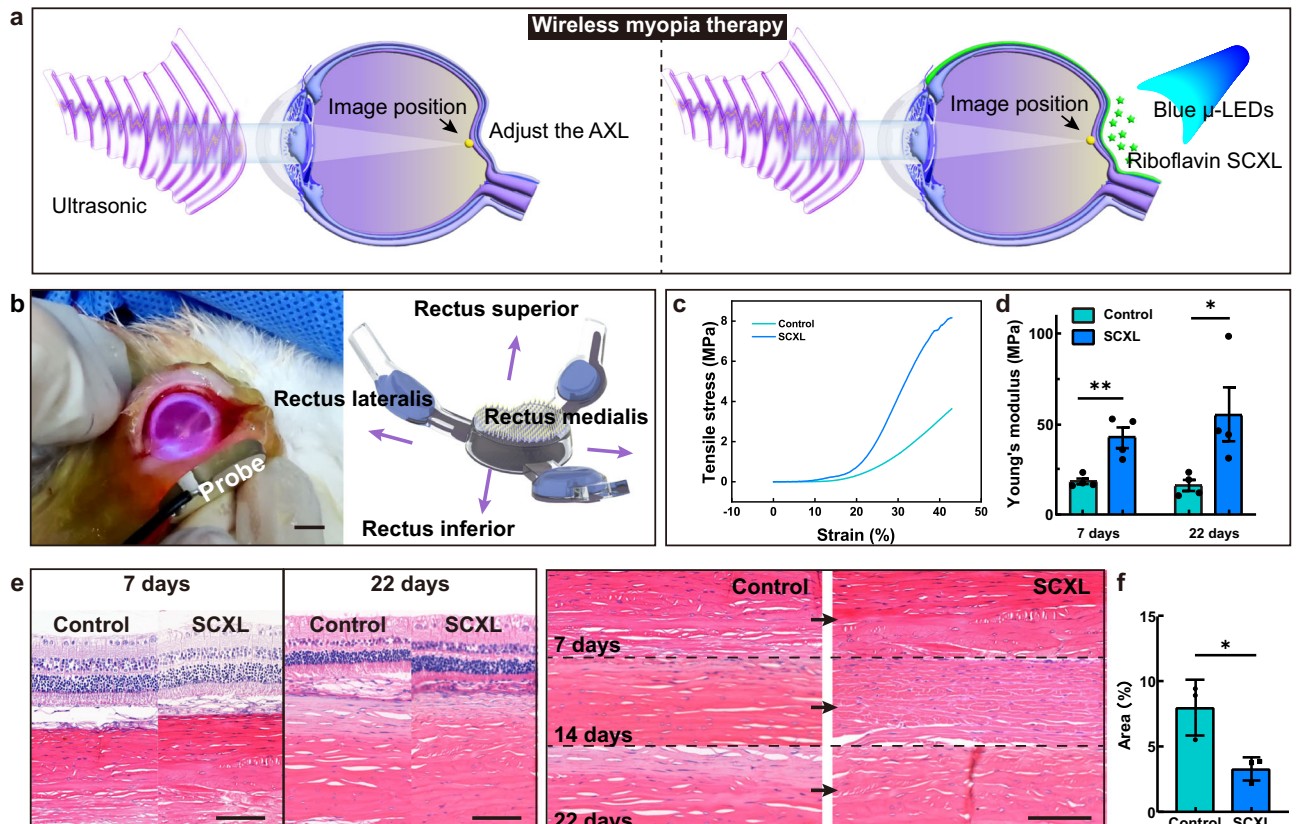

**Fig. 6 | Collagen cross-linking (SCXL) on rabbit sclera in vivo. a** Wireless ultrasonic energy system for photo-induced SCXL of the sclera. Riboflavin drugs appear as green particles in the visualization. **b** Optical images of the SCXL process. Right inset: The placement of the patch for implantation. Scale bar: 5 mm. **c** Tensile stress of rabbit eyes in the control and SCXL groups. **d** Young's modulus of rabbit eyes in the control and SCXL groups. Statistical comparisons were performed using 95% confidence interval, two-sided, unpaired $t$ test; $*p = 0.0434$, $**p = 0.0059$. $n = 4$ eyes per group. Data are presented as mean values ± SD. **e** H&E staining of the rabbit eye sections in both the control and SCXL groups. Scale bar: 100 μm. **f** Comparative analysis of tissue porosity, revealing slightly lower porosity in the SCXL group compared to the control group. Data are presented as mean values ± SD, using 95% confidence interval, two-sided, paired $t$ test; $*p = 0.0220$. $n = 3$ eyes per group. Source data are provided as a Source Data file.

The results of allograft inflammatory factor 1 (IBA 1) (Supplementary Fig. 24) staining in the SCXL group and control group revealed the presence of resting microglial cells in the inner retinal layers, with no observed activation of microglia. Immunostaining for glial fibrillary acidic protein (GFAP) demonstrates that astrocytes show a negative response, indicating that electrode and cross-linking procedures do not have any detrimental effects on the tissue. Semithin sections stained with toluidine blue (Supplementary Fig. 25) from the control group and SCXL group exhibit typical appearances of the retina, choroid, and sclera. No structural degradation or loss is observed, suggesting that the implanted patch does not result in any complications after treatment. Furthermore, no significant inflammatory reactions or damage are detected. We utilize the TUNEL (terminal deoxynucleotidyl transferase–mediated deoxyuridine triphosphate nick end labeling) reaction to detect apoptotic cells, which identifies double-stranded DNA breaks and nicks (Supplementary Fig. 26). In comparison to the control group, the SCXL group exhibits a minimal presence of tetramethyl rhodamine (TMR) red-positive cells, as indicated by arrows. Fundus photography results indicate the fundus maintains transparency, crucial for optimal vision, and indicates no complications like edema or opacification. Supplementary Fig. 27 confirms no signs of choroidal hemorrhage, retinal hemorrhage, or retinal detachment in the treated eyes, suggesting the surgery doesn't harm the fundus's structure. Angiography reveals no pathological fluorescein leakages, neovascularization, aneurysms, or capillary non-perfusion. Supplementary Fig. 28a illustrates a slight fluctuation in body weight, during the 22-day period following surgery.

Supplementary Fig. 28b displays slight variations in body temperature. All values fluctuate within the normal range. Additionally, on the 22 days post-surgery, the wounds in the therapy eye display a healing progress (Supplementary Fig. 29) comparable to that observed in the control eye, without any apparent damage. This study proves that the patch effectively decreases axial length and supports the posterior sclera. The remaining data demonstrate its safety after 22 days of implantation, showing no risk of glaucoma, cataract, retinal detachment, or endophthalmitis. Additionally, there are no obvious microscopic pathological changes in the morphology and anatomical structure of the retina.

## Discussion

This study focuses on three key areas: wireless ultrasonic modulation, high myopia therapy, and sustained release of riboflavin microneedle drugs for the sclera. Wireless ultrasound control of the eye modulation patch, compared to traditional ophthalmic surgery, offers several benefits, including less invasive extracorporeal manipulation therapy. This reduces the need for intraoperative exposure, shortens surgery time, and simplifies the overall treatment. The technique enables precise AXL regulation and SCXL performance via ultrasound, enhancing treatment accuracy and reducing the need for repeat surgeries. The study achieved an average axial length adjustment of approximately 1.217 mm in rabbit eyes, potentially translating to a significant myopia correction in humans (-2.5 diopters)[64,65]. Despite correcting refractive errors, myopia can recur due to scleral thinning and extension. Our innovative approach combines riboflavin microneedles/blue μ-LED

treatment for scleral cross-linking, resulting in a notable increase in scleral strength by about 387%. However, the study's small sample size and the relatively short 22-day post-surgery observation period mean that long-term outcomes and potential complications are yet to be fully understood. Therefore, further research and extended observation are necessary for a comprehensive assessment of the treatment's long-term safety and effectiveness. The promising preliminary results in rabbits provide a solid foundation for future human applications. This study provides valuable guidance and serves as a foundation for larger-scale experiments and the optimization of patch treatment in the future.

## Methods

### Ethical statement
These studies adhered to the ARVO Statement for the Use of Animals in Ophthalmic and Vision Research and were approved by the Animal Care and Use Committee of Sichuan Provincial People's Hospital (IRB (Research) No. 2022-245).

### Materials
The PZT was obtained from SCH Technology Co. Ltd. The 443 nm μ-LEDs were obtained from Shenzhen Rico Photoelectric Technology Co., Ltd. The red μ-LEDs were obtained from Shenzhen Super high-brightness Electronics Co., Ltd. The following reagents and antibodies were used in the study: Anti- IBA 1 Mouse mAb (Servicebio, GB12105; diluted: 1:100), Cy3 conjugated Goat Anti-mouse IgG (H + L) (Servicebio, GB21301; diluted: 1:100), Anti- GFAP Rabbit pAb (Servicebio, GB11096; diluted: 1:100), fluorescein isothiocyanate (FITC)–conjugated goat anti-rabbit IgG secondary antibody (Servicebio, GB22303; diluted: 1:100), TMR (Red) Tunel Cell Apoptosis Detection Kit (Servicebio, G1502-50T; Recombinant TdT enzyme: TMR-5-dUTP Labeling Mix: Equilibration Buffer (1: 5: 50)), Hematoxylin (Servicebio, G1004), Eosin Y (Biobomei, YE2080), 4′,6-diamidino-2-phenylindole (Servicebio, G1012), Toluidine blue (Servicebio, G1032).

### Fabrication of the eye modulation patch
The fabrication process of the eye modulation patch was schematically shown in Supplementary Fig. 30. A flexible sheet of copper-clad PI (Cu/PI/Cu, 12 μm/12.5 μm /12 μm) served as the substrate (Supplementary Fig. 30a). By utilizing the exposure, development, and etching steps, patterned transmission and interdigitated electrodes were achieved on the substrate (Supplementary Fig. 30b). Holes (diameter, 150 μm) were drilled through the substrate, and the inner walls were electroplated with copper to ensure electrical connection between the top and bottom electrodes. The interdigitated electrodes were then coated with a 75 nm layer of gold using electroless plating to prevent the oxidation of copper in the presence of NaOH solution. Critical electronic components, including a capacitor, μ-LED, and diode, were assembled onto the substrate by soldering (Supplementary Fig. 30c). A silicone mold (hardness: 30) prepared by a 3D printer was then used to fabricate the PDMS encapsulation layer and the micro-actuator solution reservoir. Before casting, mold release agent was sprayed onto the mold surface. Then, PDMS: curing (SYLGARD184, Dow Corning, USA) agent was mixed at a mass ratio of 10:1 and poured into the mold, followed by placing the flexible circuit board inside. After curing at 70 °C for 3 h and demolding, the PDMS film covering the interdigitated electrodes was removed, and a horseshoe-shaped thin film (PDMS/SBS) corresponding to the cavity shape was used to seal the micro-actuator (Supplementary Fig. 30d). Subsequently, a NaOH solution (50 mmol L$^{-1}$) (Shanghai, Macklin Biochemical Co., Ltd. Shanghai, China) was injected into the solution reservoir (Supplementary Fig. 30e), which was then sealed with a PDMS adhesive to prevent electrolyte leakage. Finally, the horseshoe-shaped microneedle array was aligned and placed on the micro-actuator and secured with water-soluble adhesive (Supplementary Fig. 30f). The orthogonal and vertical views of the silicone mold were shown in Supplementary Fig. 31a, b.

### Fabrication and characterization of PDMS/SBS membrane
The PDMS/SBS bilayer film enhanced the sealing performance of the micro-actuator solution reservoir[27]. The untreated PDMS exhibited hydrophobicity and low adhesion as shown in Supplementary Fig. 32a, at a contact angle of 115.3°. Therefore, it was necessary to apply oxygen plasma treatment on the PDMS film surface to increase its hydrophilicity and to improve its adhesion to the SBS film. The oxygen plasma modification was conducted with a constant system pressure of $3 \times 10^{-3}$ Pa, an oxygen flow rate of 40 sccm, a radio frequency plasma power of 150 W, and a plasma exposure time (treatment time) of 5 min. To investigate the hydrophobic recovery of the sample surface, static contact angles were measured using a manual contact angle goniometer and the sessile drop method with de-ionized water. A 5 μL droplet of de-ionized water was carefully placed on the sample surface using a calibrated syringe, and contact angles were measured. This approach allowed for a precise and reliable assessment of surface wettability and hydrophobicity. As demonstrated in Supplementary Fig. 32b, the water contact angle of the plasma-treated PDMS film surface was reduced to 6°. Subsequently, an SBS solution was prepared by dissolving SBS (Sigma-Aldrich, Co., USA) in toluene (Chengdu Chron Chemical Co., Ltd. China) (1/10 mL), and then the solution was dropped onto the plasma-treated PDMS film surface. The resulting film was annealed at 60 °C for approximately 3 h, followed by overnight annealing at 95 °C. The lower water contact angle of 80.2° on the surface of SBS, as shown in Supplementary Fig. 32c, facilitates adhesion between SBS and the PDMS film.

Attenuated total reflection fourier transform infrared spectroscopy (ATR-FTIR) was used to study the peak changes of the original PDMS, plasma-treated PDMS, and SBS films, as shown in Supplementary Fig. 32d. The surfaces of the PDMS samples were analyzed within 5 min after plasma treatment. A broad peak at 3420 cm$^{-1}$ appeared after plasma treatment compared to the ATR-FTIR of bare PDMS, and SBS also exhibited a low-amplitude broad peak at this position. Moreover, the peaks at 2963 and 2905 cm$^{-1}$ were more pronounced than those of bare PDMS. The new peak corresponded to hydroxyl groups that formed on the PDMS surface during plasma treatment, indicating that plasma treatment can modify the PDMS surface by adding hydroxyl groups. Therefore, it enhanced the adhesion between PDMS and SBS. The transmittance of the original PDMS, plasma-treated PDMS, and SBS films were shown in Supplementary Fig. 32e, with values of 94.29%, 88.59%, and 79.71% at 443 nm, respectively. This indicated that oxygen plasma treatment did not cause a significant loss of blue light conduction during collagen cross-linking.

The topography of pristine PDMS, PDMS/SBS, and plasma-treated PDMS samples were analyzed using an acceleration potential of 5 kV. Before imaging, all samples were sputter coated with a 10 nm gold layer. The SEM images (Supplementary Fig. 32f–k) showed that the surfaces of all samples remained smooth and intact, without any cracks or gaps, indicating good sealing and uniform deformation performance of the films before and after treatment. The cross-sectional image of the PDMS/SBS film revealed an overall thickness of approximately 440 μm, with a PDMS thickness of ~390 μm and an SBS thickness of ~50 μm, as shown in Supplementary Fig. 32l.

### Fabrication of the microneedle array
To prepare the microneedle array, a positive mold made of monocrystalline silicon was used, as shown in Supplementary Fig. 33a, b. The mold had a height of 400 μm and a bottom diameter of 370 μm. A liquid mixture of polyvinylpyrrolidone (PVP, M$_W$ = 58 kDa) (Shanghai Macklin Biochemical Co., Ltd. China) and 10 wt% riboflavin (a single dose of approximately 4.8 mg of riboflavin) (Shanghai Aladdin Biochemical Technology Co., Ltd. China) was meticulously and thoroughly stirred at room temperature. Subsequently, the mixture was poured onto a PDMS mold obtained through the process of film coating and demolding from a positive mold. The mixture was vacuum

dried at room temperature, and the resulting microneedle array was gently peeled off the mold.

## Mechanical testing of the microneedle array

The riboflavin microneedle array underwent standard mechanical tests utilizing a TMS-Pro Texture Analyser in compression mode, following previously established procedures. Subsequently, the microneedle array was meticulously positioned on the flat stainless-steel baseplate of the Texture Analyser, ensuring that the needles were oriented upwards. A flat-faced probe with a diameter of 10.0 mm descended at a controlled velocity of 0.5 mm s$^{-1}$, triggered by a force of 5 g (approximately 0.05 N). Upon contact with the microneedle array, the probe maintained its velocity of 0.5 mm s$^{-1}$ until the desired force was applied. Once the target force was achieved, the probe ascended at a speed of 0.5 mm s$^{-1}$. To evaluate the compression strength, graphs depicting the relationship between compression force and displacement were generated.

## Characterization and measurements

Utilized Sweep source optical coherence tomograph and fundus angiography system (BM-400K BMizar, Toward Pi Medical Technology, Beijing, China) and Panoramic ophthalmoscope (Daytona (P200T), OPTOS PLC, United Kingdom) for comprehensive fundus imaging and observation. The investigation of materials' structure and morphology was carried out through Scanning Electron Microscopy (SEM, GeminiSEM 300, Germany). Optical biometry (IOL Master 700, Carl Zeiss Meditec AG, Jena, Germany) and Ocular ultrasonography (A-B Scan, MD-2300S, China) were utilized for measuring axial length. The intraocular pressure was measured using the Contact Rebound Tonometer IOP mini (Icare TONOVET Plus, United State). Ophthalmic camera (DX100-01A, China and DX100-01A, China) were angiography and fundus photography. Transmission Electron Microscope (JEM-1400-FLASH, Japan) was used to observe the morphology. Digital slice scanner (Pannoramic 250, China) for H&E and toluidine blue slice analysis. To create precise patch molds, a 3D Printer (ZRapid ISLA660, China) was utilized. For the analysis of material surface functional groups, a Fourier Infrared Spectrometer (FT-IR) Spectrometer (INVENIO, Germany) was employed. To test the mechanical strength of the microneedles, a TMS-Pro Texture Analyzer was utilized (Food Technology Corporation, USA).

## Animals testing

Conventional grade New Zealand white rabbits, comprising five males and five females, aged three months and weighing between 2.5–3.0 kg, were utilized in this study. All procedures were conducted in strict accordance with the approved study protocols and relevant regulatory guidelines. All experimental rabbits were purchased from the Biotechnology Corporation of Dashuo (Chengdu, China). The rabbits were maintained under 12-h light/12-h dark conditions. Rabbits with any form of lesion in the cornea, lens or fundus were excluded.

## Groups

Ten New Zealand White rabbits were enlisted for this research. Patch implantation was exclusively performed in the rabbits' right eyes, while no interventions were undertaken on their corresponding left eyes. To reduce the impact of individual variability, the right eye of the treatment group functioned as both the control and experimental eye before the implantation (pre-treatment) patch and following the treatment (post-treatment), during the axial length adjustment surgery.

## Surgery, modulation of axis oculi and cross-linking treatment

The retinal surgery team, consisting of skilled specialists, performed all surgeries. For the induction of general anesthesia, 1 mg/kg of 3% sodium pentobarbital (Beijing Chemical Reagent Co. Beijing, China) was injected into the auricular veins of the experimental rabbits. Prior

to surgery, the intraocular pressure, axial length, OCT and fundus images of both eyes of all rabbits were measured. After data collection, a 5% povidone iodine solution (Chengdu Yongan Pharmaceutical Co. Ltd., Chengdu, China) was used to sterilize the instrument. Next, the rabbit was placed on the operating table with the right eye upward, and routine disinfection was performed with an eyelid opener. A 360° conjunctival incision was made on the right eye of all rabbits, and the extraocular muscles and rectus muscles were separated to fully expose the posterior sclera of the eyeball. The patch was implanted into the right eye, and the position of the patch was adjusted. The fundus was observed through binocular indirect ophthalmoscopy. The appropriate position of the apex pressure point was within a range of 5–8 mm below the optic disc and the horizontal seam. Upon identifying the appropriate experimental site, the patch was carefully positioned at the designated location and pressed against the exposed sclera. Once accurately placed, the three legs of the patch were securely sutured to the sclera (Supplementary Fig. 34) using a 5-0 Coated Vicryl Plus synthetic absorbable suture (Ethicon, Inc. USA). Following the implantation of the patch, the conjunctiva was then sutured to complete the procedure. The patch was allowed to remain still for 30 min to ensure full infiltration of riboflavin into the sclera. The patch was activated to initiate the electrolysis function, targeting the modulation of the axial length on the posterior sclera. This process lasted for 6 min. Subsequently, blue μ-LEDs were employed to irradiate the sclera for 30 min, incorporating a 1-min break every 5 min. The objective was to achieve complete cross-linking between the riboflavin and the sclera. After cross-linking, the intraocular pressure and axial length of the left and right eyes were measured again. The intraocular pressure, axial length, OCT and fundus images of both eyes of all rabbits were measured again. After the surgery, the therapy eyes were treated with tobramycin and dexamethasone eye ointment (TobraDex, ALCON CUSI S.A. Spain) to prevent infection. The rabbits were closely monitored until they regained consciousness before being returned to the animal room. The vital signs and eye conditions of the experimental rabbits were observed daily, and body weight and temperature were measured.

## H&E staining, immunofluorescence assay and Tunel assay

The rabbits were euthanized using an overdose of pentobarbital, and their eyes were enucleated for histological examination. After enucleation, all eyes were fixed in 2% paraformaldehyde and 2.5% glutaraldehyde solution, dehydrated in a series of increasing alcohol concentrations, and embedded in paraffin. Sections cut at a thickness of 4 μm were stained with Hematoxylin (G1004; Servicebio; China), Eosin Y (YE2080; Biobomei, China) (H&E) and Toluidine blue (G1032, Servicebio; China) and examined under a light microscope.

Sections of samples embedded in paraffin were deparaffinized in a xylene ethanol series, placed in Tris-EDTA buffer for antigen retrieval (10 mM Tris, 1 mM EDTA, 0.05% Tween, pH = 9.0). The sections were then blocked using 5% bovine serum albumin. For immunostaining, sections underwent treatment with IBA 1 primary antibody (GB12105; Servicebio, China) at a dilution of 1:100 using the primary antibody dilution buffer (G2025, Servicebio). Detection of the primary antibodies was achieved with a Cy3-conjugated Goat Anti-mouse IgG (H + L) secondary antibody (GB21301; Servicebio, China), also diluted at 1:100 in PBS (G0002; Servicebio, China). In addition, sections were stained for GFAP using a primary antibody (GB11096; Servicebio, China) at a 1:100 dilution with the primary antibody dilution buffer (G2025, Servicebio). This was detected using a fluorescein isothiocyanate (FITC)−conjugated goat anti-rabbit IgG secondary antibody (GB22303, Servicebio, China), diluted 1:100 in PBS (G0002; Servicebio, China). Nuclei staining was conducted with 4′,6′-diamino-2-phenylindole (DAPI) (G1012, Servicebio, China) without dilution.

For the detection of cell apoptosis using the TMR (Red) Tunel Cell Apoptosis Detection Kit (Servicebio, G1502-50T), the kit components

are utilized to prepare a TdT incubation buffer. This preparation involves combining Recombinant TdT enzyme, TMR-5-DUTP labeling mix, and equilibration buffer in a 1:5:50 volume ratio. Following this, the samples are labeled, allowing for the identification of apoptotic cells. DAPI staining is employed to visualize all cells in blue, whereas apoptotic nuclei incorporate TMR-5-DUTP, resulting in the manifestation of red fluorescence.

## Transmission electron microscopy (TEM) observation

The tissue samples were first fixed in 3% glutaraldehyde for primary structural stabilization, followed by postfixation in 1% osmium tetroxide to enhance preservation and contrast. Subsequent dehydration was achieved through a graded series of acetone solutions, essential for complete water removal before resin infiltration. The tissues were then thoroughly infiltrated with Epox 812 resin over an extended period, ensuring deep resin penetration, and embedded to form solid blocks for sectioning. Semi-thin sections were stained with methylene blue for preliminary light microscopic examination. For detailed ultrastructural analysis, ultrathin sections were prepared using a diamond knife, stained with uranyl acetate and lead citrate to improve electron density and contrast, and then examined under a JEM-1400-FLASH Transmission Electron Microscope.

## Statistics and reproducibility

All data showed the means ± standard deviation (SD) of at least three biological replicates with the n indicated in each experiment. The statistical analyses were indicated in the legends of each figure, with $p < 0.05$ indicating a statistically significant difference. a two-tailed Student $t$ test was performed for two group comparisons. One-way ANOVA analysis of variance was used to determine the statistically significant difference for multiple group comparisons. The statistical analysis was performed in Prism 8 (GraphPad Software, San Diego, CA, USA). For fluorescence imaging, hematoxylin and eosin (H&E) staining, immunofluorescence analysis, and TUNEL apoptosis assays, the experiments were independently replicated three times, yielding similar results.

## Reporting summary

Further information on research design is available in the Nature Portfolio Reporting Summary linked to this article.

## Data availability

All data supporting the findings of this study are available within the article and its supplementary files. Any additional requests for information can be directed to, and will be fulfilled by, the corresponding authors. Source data are provided with this paper. The source data is available via Zenodo at https://doi.org/10.5281/zenodo.10619661. Source data are provided with this paper.

## Code availability

The code for data analysis and figure generation related to sound pressure simulation is available via Zenodo at https://doi.org/10.5281/zenodo.10592685.

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

## Acknowledgements

We thank Zheng Lin Yang of the Department of Ophthalmology, Sichuan Provincial People's Hospital for discussion on data presentation. This work was supported by National Natural Science Foundation of China (11674048, L.X.), Sichuan Science and Technology Program (2020JDJQ0026, X.X. and 2021YFG0140, L.X.), Sichuan Provincial Cadre Health Research Project (ZH2024-201, J.Z.), Sichuan Province Central Government Guides Local Science and Technology Development Special Project (2021ZYD0108, J.Z.) and Radiation Oncology Key Laboratory of Sichuan Province Open Fund (2021ROKF01, X.X. and 2022ROKF02, L.X.).

## Author contributions

T.Z., X.X. and J.Z. put forward the concept. T.Z., H.Y. and J.L. carried out the experiment. T.Z., H.Y., J.G., J.L., M.L., X.G., S.C. and Q.H. conducted the investigation. T.Z., H.Y., J.G., J.L., H.G., S.L., R.L., Z.L. and Y.W. carried out the analysis and visualized the data. J.Z., X.X., L.X., Yang Z. and Yan Z. supervised the work. T.Z. wrote the manuscript. J.Z., X.X., L.X., Yang Z., Yan Z. and C.S. reviewed and edited the manuscript.

## Competing interests

The authors declare no competing interests.

## Additional information

[1]School of Physics, University of Electronic Science and Technology of China, Chengdu 611731, China. [2]School of Medicine, University of Electronic Science and Technology of China, Chengdu 610054, China. [3]Department of Ophthalmology, Sichuan Provincial People's Hospital, University of Electronic Science and Technology of China, Chengdu, China. [4]School of Information and Communication Engineering, University of Electronic Science and Technology of China, Chengdu 611731, China. [5]Department of Mechanical Engineering, City University of Hong Kong, Hong Kong SAR 999077, China. [6]Brain Cognition and Brain Disease Institute, Shenzhen Institutes of Advanced Technology, Chinese Academy of Sciences, Shenzhen 518055, China. ✉ e-mail: zhongjie@med.uestc.edu.cn; xuexinyu@uestc.edu.cn

