## [Peer Review File · Nature Communications]

REVIEWER COMMENTS

Reviewer #1 (Remarks to the Author):

The authors proposed an implantable patch to correct for Myopia Therapy. the patch is capable of deforming the posterior pole of the eye to reduce the AXL.

While the goal of the study seems to be achieved and the data are technically sound, it remains unclear the relevance of the study. Authors proposed an interesting tech device. However, the authors failed to convince that there is real clinical need behind the device. Yes, myopia is a common condition. As the authors wrote, there are plenty solutions to compensate for it, including glasses and lenses which are noninvasive. Not to mention LASIK, ICL and IOL. The authors justify their design by mentioning that none of these approaches address the root cause (axial length) but just compensate for it. In my opinion, the goal is to correct for refractive errors, not necessarily addressing the root cause; also, the less invasive the better. Authors must clearly explain the relevance in their approach for clinical use. Otherwise it remains a nice tech but with no real use.

Detailed comments

1. While reading i was wondering about IOP. I found that you did not address this point appropriately. Figure 5k revealed a remarkable increase in IOP post surgery. This is a problem considering the risk of inducing glaucoma just to correct a refractive error. Did you measure IOP over time during the 7 days post implantation to check if it returns to normal level? I think it is mandatory to control IOP otherwise the risk of glaucoma outweighs any benefits in refractive change
2. It appears to me that the device is left implanted for 7 days. If correct, why? Can't you remove it immediately after surgery? Or is it required to keep the eye deformed.
3. Is the process reversible? After CXL, can the process be reversed for explanation or simplify to reduce IOP?
4. Data are presented as mean±s.e.m. Why? Statistical representation of distributions requires mean±s.d.

Reviewer #2 (Remarks to the Author):

This manuscript describes a multifunctional therapeutic patch for myopia vision correction and relapse prevention. The therapeutic patch was designed to contain piezoelectric transducers, an electrochemical

micro-actuator, a drug microneedle array, μ -LEDs, and a flexible circuit with biocompatible encapsulation in a compact and wireless system. Placed near the optic nerve, the patch converted external ultrasound into electrical energy, triggering the electrochemical micro-actuator to shorten eye's axial length (AXL). Simultaneously, the drug microneedle array delivered riboflavin to the sclera and the blue light from μ -LEDs crosslinked the sclera, preventing myopia-induced relaxation. In vivo experiments on rabbits confirmed the feasibility for this innovative approach, offering the potential benefits for myopia treatment and prevention. This manuscript seems interesting, but should be revised in many parts as commented below before publication.

Major issue

[1] In general, myopia vision correction has been performed by the modulation of cornea. For examples, Lasik surgery is carried out by the modification of cornea with laser and keratoconus is treated by the crosslinking with riboflavin under UV irradiation. In this context, authors should discuss the rationale and the possible negative side-effect of this work with the modification of sclera instead of cornea.

[2] In page 10, line 436, authors described the potential for individual rabbit variations and the influence of surgical sutures on patch implantation adjustments, requiring the secondary electrolysis session to achieve the optimal length. Authors should clearly explain the effect of electrolysis conditions such as voltage and time on AXL changes to make precise adjustments. In addition, authors should describe the AXL length for hyperopia, myopia, and normal vision.

[3] In Figure 4J, authors should explain the extent to which axial elongation occurs according to Young's modulus after cross-linking (CXL).

[4] In Figure 5K, IOP increased by 6.1 mm/Hg. Authors should discuss the possible negative side effect such as glaucoma by the increase in IOP.

[5] How was the patch removed? If the patch remained, it might cause the negative side effect by the immune response. The patch size seems too large to be biocompatible.

Minor issue

[1] In Figure 4H, authors should provide the scale bar.

[2] Authors should indicate the molecular weight of PVP used for microneedles in the section of Materials and Method.

[3] Authors should describe the size and thickness of the patch.

POINT-BY-POINT REPLY TO PEER REVIEW COMMENTS

(Manuscript Number: NCOMMS-23-37475A-Z)

REVIEWER #1 COMMENTS:

The authors proposed an implantable patch to correct for Myopia Therapy. The patch is capable of deforming the posterior pole of the eye to reduce the AXL.

While the goal of the study seems to be achieved and the data are technically sound, it remains unclear the relevance of the study. Authors proposed an interesting tech device. However, the authors failed to convince that there is real clinical need behind the device. Yes, myopia is a common condition. As the authors wrote, there are plenty solutions to compensate for it, including glasses and lenses which are noninvasive. Not to mention LASIK, ICL and IOL. The authors justify their design by mentioning that none of these approaches address the root cause (axial length) but just compensate for it. In my opinion, the goal is to correct for refractive errors, not necessarily addressing the root cause; also, the less invasive the better. Authors must clearly explain the relevance in their approach for clinical use. Otherwise, it remains a nice tech but with no real use.

REPLY TO REVIEWER #1:

Authors: We thank the Reviewer for the very useful comments on the aspects of the clinical significance. Below we show our point-by-point replies.

1. Clinical Need for the Patch:

The reviewer's insightful suggestions have led to a reevaluation of our initial definition, highlighting that our patch is more aptly suited for application in the field of high myopia (≤ -5.0 D). This clarification is significant, as high myopia substantially increases the risk of developing more severe forms of the condition, which can potentially lead to blindness [Morgan, I. G. et al. *Lancet* **379**, 1739-1748 (2012)]. Presently, common myopia management strategies, such as glasses, LASIK, and corneal refractive surgery, are primarily focused on correcting refractive errors. However, these methods fall short in addressing the axial elongation of the eye, a key factor in the progression of myopia [Kim, T. I. et al. *Lancet* **393**, 2085-2098 (2019)]. This oversight is particularly critical in cases of high myopia, where ongoing axial elongation poses a risk for severe and vision-threatening complications. Individuals with high myopia are susceptible to serious ocular deformities and pathologies, including posterior staphyloma, lacquer cracks, and myopic choroidal neovascularization, each of which has the potential to cause irreversible vision loss if not properly managed. Consequently, for early and advanced cases of high myopia, invasive therapeutic interventions are often necessary to address these complications effectively [Daniel Ian Flitcroft et al. *Invest. Ophthalmol. Vis. Sci.* **60**, M20-M30

(2019)].

2. Rationalizing the Use of the Patch:

The innovation of our implantable patch lies in its capacity to deform the posterior pole of the eye, thereby effectively managing the axial elongation. Additionally, our patch utilizes collagen cross-linking treatment to enhance scleral rigidity. This dual approach not only aims to correct the existing axial elongation but also to strengthen the sclera against further deformation. We believe this offers a preventative strategy, particularly valuable in the management of progressive high myopia and pathological myopia, potentially mitigating the risk of severe ocular pathologies associated with axial elongation. This approach goes beyond mere refractive correction and targets the structural cause of myopia.

(1) Therapeutic Advantages for High Myopia:

The patch is specifically designed to target and reduce the elongation of the eye's axial length, a fundamental structural issue in high myopia. High myopia often leads to a weakened and thinned sclera, which can exacerbate the elongation of the eyeball. The patch works to reinforce the sclera, enhancing its biomechanical properties and preventing further deformation. By applying the patch in the early stages of high myopia, it acts as a preventive solution to halt or significantly slow down the progression of the condition, thereby reducing the risk of developing more severe myopic complications. The design of the patch for treating progressive high myopia incorporates a feature that allows for its immediate removal post-treatment, as shown in **Fig.R1**.

(2) Therapeutic Advantages for Pathological Myopia:

High myopia carries a substantial risk of progression to pathological myopia if left uncontrolled. Pathological myopia can lead to severe ocular conditions, notably myopic maculopathy and myopic traction maculopathy. The development of a specialized patch, designed to cover the posterior pole of the eye, offers a proactive solution in managing these risks. This patch provides continuous support to the eye, specifically targeting the inward contraction of the relaxed posterior pole structure. It exerts a controlled force to maintain the structural integrity of the eye. The patch extends beyond basic vision correction, playing a pivotal role in posterior scleral reinforcement to mitigate complications associated with high and pathological myopia [Zhu, Z. et al. *Clin. Exp. Ophthalmol.* **37**, 660-663 (2009); Zhu, S. Q. et al. *Br. J. Ophthalmol.* **102**, 1701-1704 (2018); Zhu, S. Q. et al. *Br. J. Ophthalmol.* **100**, 1470-1475 (2016)]. It shows potential in reducing vitreous body traction on the macula. This reduction could significantly lower the risk of retinal detachment, a common and serious complication in advanced myopia. Furthermore, the patch may help in preventing choroidal atrophy, a critical issue in pathological myopia that can lead to further vision deterioration. By addressing these structural issues at their source, the treatment aims to halt the progression of the disease, potentially leading to improved visual outcomes.

In conclusion, the eye conditioning patch stands out as a comprehensive solution for both high and pathological myopia, addressing specific needs and complications associated with each stage.

Fig. R1 | Depiction of patch treatment in high myopia progression.

To accommodate the comments, we've updated the discussion in our revised manuscript (line 35, page 1) with further details in **Supporting Fig.1**.

Comment 1. While reading I was wondering about IOP. I found that you did not address this point appropriately. Figure 5k revealed a remarkable increase in IOP post-surgery. This is a problem considering the risk of inducing glaucoma just to correct a refractive error. Did you measure IOP over time during the 7 days post implantation to check if it returns to normal level? I think it is mandatory to control IOP otherwise the risk of glaucoma outweighs any benefits in refractive change.

Reply 1. We express our gratitude to the Reviewer for their insightful observations and the valuable suggestion to monitor intraocular pressure (IOP) in our study. We concur that vigilant monitoring of IOP post-operatively is crucial for preventing potential glaucoma development due to elevated intraocular pressure. In response to this suggestion, we have augmented our data with continuous IOP monitoring over a period of 22 days. During this extended monitoring period, we observed an initial increase in IOP in the six rabbits subjected to surgery, which we attribute to the surgical procedure itself and the subsequent implantation of the patch. This elevation in IOP was marked and reached a peak on the day following the surgery, as shown in **Fig. R2**. It's notable that the maximum post-surgery intraocular pressure was around 23.6 mmHg, significantly lower than the average intraocular pressure seen in glaucoma cases (**Table R1**).

Fig. R2 | Monitoring of intraocular pressure (IOP) changes over a 22-day period: (0-) OD before surgery, (0+) OD after surgery (Patch retained group).

It is important to emphasize that, following the initial post-surgical elevation, the intraocular pressure (IOP) values began to normalize by the third day after the surgery. During the comprehensive 22-day monitoring period, we diligently recorded the IOP readings (n = 6). We observed a stabilization of IOP values close to preoperative levels starting from the third day post-surgery, with this stability maintained consistently up to the 20th day. This trend is a positive indication, suggesting that the initial surge in IOP observed post-surgery is a temporary response rather than a lasting complication. In terms of specific values, the highest recorded IOP was 23.6 mmHg, while the lowest was 8.2 mmHg. Generally, the IOP tended to fluctuate within a range of 8.3 to 12.6 mmHg. These detailed findings, including the range and stabilization of IOP values post-surgery, are presented in the revised manuscript (see Page 10, Line 435) and are further illustrated in Supplementary Figure 23. This data provides valuable insights into the transient nature of IOP changes following surgical intervention and patch implantation, reinforcing the potential safety and efficacy of this treatment method.

Table R1 | Intraocular pressure reference value.

IOP (mmHg)	This research	Glaucoma	Hypotony
Control	9.0-13.8		
Post-implantation (0+)	16.6-23.6	30-37 ¹	$\leq 6^2$
22 days	8.3-12.6		

1. Ueno, A., Tawara, A., Kubota, T. et al. Histopathological changes in iridocorneal

angle of inherited glaucoma in rabbits. *Graefe's Arch. Clin. Exp. Ophthalmol.* **237**, 654–660 (1999).

2. Adan V., Nikos S., Stéphane B., André M., Alexandre M., Sylvain R. In vivo testing of a novel adjustable glaucoma drainage device. *Invest. Ophthalmol. Vis. Sci.* **55**, 7520-7524 (2014).

Comment 2. It appears to me that the device is left implanted for 7 days. If correct, why? Can't you remove it immediately after surgery? Or is it required to keep the eye deformed.

Reply 2. We appreciate the reviewer inputs regarding the removability of the patch, prompting us to clarify the application of the patch in patients with high myopia. In cases where high myopia continues to progress, leading to excessive elongation of the eye axis without associated fundus lesions, the patch can be strategically utilized. Its role in these scenarios is to perform scleral collagen cross-linking, thereby strengthening the posterior pole sclera. This intervention is crucial in preventing the thinning and relaxation of scleral tissue caused by the elongated eye axis, thus averting the progression towards pathological myopia. For patients experiencing only progressive high myopia without pathological changes, the device is designed to be removable immediately after the completion of postoperative collagen cross-linking. This approach is suitable when the primary objective is to surgically reinforce the sclera, without the presence of pathological myopia.

We explored the immediate removal of the patch following the scleral collagen cross-linking procedure (n = 5). After surgery (0+), the peak IOP was recorded at 28.3 mmHg (**Fig. R3**), importantly staying below the threshold typically associated with an increased risk of glaucoma (**Table R1**). A noteworthy decrease in IOP was observed by the second day post-surgery, suggesting a prompt and positive response to the treatment. By the third day, IOP levels had almost normalized to preoperative values, demonstrating both a quick recovery and stable stabilization of eye pressure. Over the course of a 22-day observation period, the IOP consistently remained within the normal range, fluctuating between 10.5 and 12.2 mmHg. The maintenance of IOP within normal parameters post-removal of the device.

These observations suggest the practicality and safety of removing the device immediately after scleral reinforcement, particularly in patients with progressive high myopia not exhibiting pathological changes.

Fig. R3 | Monitoring of intraocular pressure (IOP) changes over a 22-day period: (0 -) OD before surgery, (0+) OD after surgery (Patch removal after sclera collagen cross-linking (SCXL)).

Following the Referee's comment, we have incorporated revisions in the manuscript (line 96, page 3).

Comment 3. Is the process reversible? After CXL, can the process be reversed for explanation or simplification to reduce IOP?

Reply 3. Riboflavin (vitamin B2) combined with blue light-induced scleral collagen crosslinking is generally considered to be an irreversible process. This technique, primarily used in ophthalmology to strengthen collagen fibers, works by inducing additional crosslinks within the collagen fibers of the eye's sclera (or cornea). When riboflavin is applied to the scleral tissue and activated by blue light, it generates reactive oxygen species. These reactive species lead to the formation of additional covalent bonds between collagen molecules. The additional crosslinks that form as a result of this process increase the biomechanical strength and stability of the scleral tissue [Mazzotta, C. et al. “[Principles of Accelerated Corneal Collagen Cross-Linking]” in *Management of Early Progressive Corneal Ectasia: Accelerated Crosslinking Principles* (Springer Nature, Switzerland, 2017), pp. 5-6; Prasad, P. N. “[Photobiology for Biophotonics]” in *Advances in Biophotonics* (IOS Press, 2005), pp. 123-147.]. This is particularly beneficial in conditions like high myopia, where the sclera may be weakened and prone to deformation. Once these new crosslinks are formed, they are generally permanent. The process fundamentally alters the structure of the collagen, creating a more rigid and stable tissue matrix.

In summary, riboflavin and blue light-induced scleral collagen crosslinking is an irreversible process, leading to permanent changes in the collagen structure of the eye's sclera. This property is harnessed therapeutically to strengthen and stabilize scleral tissue.

The continuous postoperative IOP monitoring results over a 22-day period provide valuable reference into the impact of the surgical procedure and subsequent recovery on IOP levels. Given that the IOP did not show a sustained increase but instead returned to normal levels relatively quickly after surgery, the return to and maintenance of preoperative IOP levels post-surgery are reassuring with regard to the risk of glaucoma (see detailed explanation in the **Reply 1**).

Following the Referee's comment, we have incorporated revisions in the manuscript (line 264, page 6).

Comment 4. Data are presented as mean \pm s.e.m. Why? Statistical representation of distributions requires mean \pm s.d.

Reply 4. Thank you for pointing these out to us. We acknowledge our oversight. According to your suggestion, the presentation of the normally distributed data in the full text has been changed to mean \pm s.d. (See Page 25, Line 915; Page 26, Line 930; Page 27, Line 944 in the revised manuscript, **Figure 4j**, **Figure 5k**, **Figure 6d** and **f**).

REVIEWER #2 COMMENTS:

This manuscript describes a multifunctional therapeutic patch for myopia vision correction and relapse prevention. The therapeutic patch was designed to contain piezoelectric transducers, an electrochemical micro-actuator, a drug microneedle array, μ -LEDs, and a flexible circuit with biocompatible encapsulation in a compact and wireless system. Placed near the optic nerve, the patch converted external ultrasound into electrical energy, triggering the electrochemical micro-actuator to shorten eye's axial length (AXL). Simultaneously, the drug microneedle array delivered riboflavin to the sclera and the blue light from μ -LEDs crosslinked the sclera, preventing myopia-induced relaxation. In vivo experiments on rabbits confirmed the feasibility for this innovative approach, offering the potential benefits for myopia treatment and prevention. This manuscript seems interesting, but should be revised in many parts as commented below before publication.

Authors: We acknowledge Reviewer #2 for their effort in assessing our manuscript. The positive feedback on our work, particularly on the innovative treatment approach using a wireless ultrasound-induced patch for modulating high myopia, is greatly appreciated. We are especially encouraged by the Reviewer's opinion that our research meets the standards for publication in *Nature Communications*. We would like to highlight that the Reviewer's comments helped enormously for the improvement of our manuscript. Below we addressed the comments point-by-point.

REPLY TO REVIEWER #2:

Major issue:

Comment 1. In general, myopia vision correction has been performed by the modulation of cornea. For examples, Lasik surgery is carried out by the modification of cornea with laser and keratoconus is treated by the crosslinking with riboflavin under UV irradiation. In this context, authors should discuss the rationale and the possible negative side-effect of this work with the modification of sclera instead of cornea.

Reply 1. The reviewer's insightful suggestions have led to a reevaluation of our initial definition, highlighting that our patch is more aptly suited for application in the field of high myopia (≤ -5.0 D). This clarification is significant, as high myopia substantially increases the risk of developing more severe forms of the condition, which can potentially lead to blindness [Morgan, I. G. et al. *Lancet* **379**, 1739-1748 (2012)]. Presently, common myopia management strategies, such as glasses, LASIK, and corneal refractive surgery, are primarily focused on correcting refractive errors. However, these methods fall short in addressing the axial elongation of the eye, a key factor in the progression of myopia [Kim, T. I. et al. *Lancet* **393**, 2085-2098 (2019)]. This oversight is particularly critical in cases of high myopia, where ongoing axial elongation poses a risk for severe and vision-threatening complications. Individuals with high myopia are susceptible to serious ocular deformities and pathologies, including posterior staphyloma, lacquer cracks, and myopic choroidal neovascularization, each of which has the potential to cause irreversible vision loss if not properly managed. Consequently, for early and advanced cases of high myopia, invasive therapeutic interventions are often necessary to address these complications effectively [Daniel Ian Flitcroft et al. *Invest. Ophthalmol. Vis. Sci.* **60**, M20-M30 (2019)]. Below, we address the rationale behind this choice and its implications:

1. Rationale for Scleral Modification in High Myopia:

High myopia often involves axial elongation of the eye, which is more directly related to the sclera than the cornea. By focusing on scleral modification, our treatment addresses this fundamental structural change in the eye. The scleral reinforcement, achieved through collagen cross-linking induced by our wireless ultrasound-induced patch, aims to prevent further elongation and weakening of the sclera, thereby halting or slowing the progression of high myopia. Scleral collagen cross-linking works basically the same way as corneal collagen cross-linking. The technique of riboflavin (vitamin B2) combined with blue light-induced scleral collagen crosslinking is employed to reinforce collagen fibers in the eye's sclera, similar to its use in the cornea. The application of riboflavin, followed by blue light activation, creates reactive oxygen species that facilitate the formation of additional covalent bonds between collagen molecules. This process enhances the biomechanical strength and stability of the sclera, counteracting the weakening and deformation often seen in high myopia. Given that high myopia typically involves axial elongation primarily related to the sclera, our treatment targets this key structural change.

While corneal treatments like LASIK effectively correct refractive errors, they do not address the axial elongation associated with high myopia. Our scleral modification

approach complements corneal techniques by targeting a different aspect of myopia progression, specifically beneficial in cases at risk of developing pathological changes due to axial elongation.

2. Possible Negative Side-Effects:

In our research, we have adopted wireless ultrasonic control technology to address some of the challenges commonly associated with traditional scleral modification surgeries. This method centers around the use of a controlled, wireless patch, designed to improve the procedure's safety and effectiveness in a measured way. A significant aspect of our approach is the controlled adjustment of the eye's axial length and scleral strength post-surgery. Traditional surgeries often require extended operative times due to the need for manual implantation, potentially increasing exposure risks. Our method aims to reduce these risks by minimizing the duration of surgical exposure. Traditional methods usually do not allow for postoperative adjustments, which can lead to challenges in precision. Wireless technology is intended to offer more control and accuracy in this aspect. The biocompatible, flexible materials can reduce the likelihood of adverse reactions and to ensure compatibility with the eye's natural structure. Postoperative analyses, including pathology and immunohistochemistry, have not indicated significant inflammation or complications. Additionally, intraocular pressure has generally remained stable. In conclusion, while our approach incorporates advancements in technology and materials, we recognize the importance of ongoing research and evaluation to fully ascertain its effectiveness and safety. Our findings so far are promising, but we continue to approach our research with diligence and an understanding of the complexities involved in ocular treatments.

We acknowledge that the observation period (22-day) following our surgical treatment has been relatively brief. This limitation means that while initial results are promising, they may not fully capture long-term outcomes or potential delayed complications. Consequently, extended observation and continued research are necessary to thoroughly assess the treatment's long-term safety and effectiveness. And our treatment has not been applied clinically. This is an important consideration, as results observed in controlled experimental settings, such as with animal models, may not directly translate to human clinical scenarios. Clinical trials will be essential to evaluate the treatment's safety and efficacy in human patients. The dimensions of our patch – 4 mm in thickness and 19 mm between the two legs – were specifically designed for ease of implantation in the scleral region at the rear pole of a rabbit's eye. Given the relatively larger size of human eyes compared to rabbits', we anticipate that the application of the device in human eyes would be feasible and potentially safer. However, this assumption needs to be rigorously tested in clinical trials to ensure compatibility and safety for human patients.

In summary, while our study presents an innovative approach to therapy high myopia, further research, including longer observation periods and clinical trials, is essential to validate its safety and effectiveness for potential human application. The promising results from our preliminary studies with rabbits provide a foundation for this future work, but they are just the first step in a comprehensive process to ensure the treatment's applicability and safety in humans.

Following the Referee's comment, we have incorporated revisions in the manuscript (line 35, page 1; line 422, page 10; line 489, page 11), and more details can be found in **Supporting Fig. S22-29**.

Comment 2. In page 10, line 436, authors described the potential for individual rabbit variations and the influence of surgical sutures on patch implantation adjustments, requiring the secondary electrolysis session to achieve the optimal length. Authors should clearly explain the effect of electrolysis conditions such as voltage and time on AXL changes to make precise adjustments. In addition, authors should describe the AXL length for hyperopia, myopia, and normal vision.

Reply 2.

Pressure in the pump chamber

To quantitatively predict the pressure in the micro-actuator and drug reservoir, a theoretical model is established for the whole micro-actuator system. Ideal gas law of the pump chamber gives

$$P(V + V_0) = nRT \quad (\text{Eq S1})$$

where P is the pressure, V_0 and V are the initial volume change of air inside the micro-actuator, respectively, n is the number of moles of air, R is the ideal gas constant, and T the temperature.

From Faraday's laws of electrolysis

$$Q = nZF \quad (\text{Eq S2})$$

where Q is the total charge passed, Z is the number of electron transfers for the product, and F is the Faraday constant.

it can be known that the number of moles of gas molecules

$$n = \frac{3it}{4F} \quad (\text{Eq S3})$$

From Eq S1 of the college, the volume of the micro-actuator V can be related by

$$V = \frac{3it}{4FP} RT \quad (\text{Eq S4})$$

The volume of deformation V generated by the chamber can be equivalently represented as the volume of a dome, which can be calculated as

$$V = \pi h \frac{(h^2 + 3a^2)}{6} \quad (\text{Eq S5})$$

where h is the height of the members deformation, and a is the radius of the chamber. The volume of a micro-actuator undergoes change when gas pressure is applied. The formula for the Bulk modulus (k) can be expressed as

$$k = \frac{E}{3(1-2\nu)} \quad (\text{Eq S6})$$

$$k = -V_0 \frac{\partial P}{\partial V} \quad (\text{Eq S7})$$

where k is Bulk modulus, E is Young's modulus and ν is Poisson ration.

According to Eq S4, Eq S5, Eq S6 and Eq S7, the functional relationship between maximum displacement of flexible membrane (h) and t can be obtained as

$$EF\pi^2h^2(3a^2 + h^2)^2 = 81iRT\pi la^2(1 - 2\nu)t \quad (\text{Eq S8})$$

The theoretical analysis yielded the formula Eq S8. We conducted a comparative analysis between our experimental results and the theoretical predictions, resulting in a notable correlation that is illustrated in Figure 3h. This comprehensive formula is instrumental in representing the relationship of the AXL variation h and the time t . The alignment of our experimental findings with this theoretical model underscores the accuracy of the model in predicting the biomechanical behavior of the scleral tissue post-treatment, demonstrating the potential applicability of wireless ultrasonic control high myopia therapy in clinical settings. The manuscript has been added to this formula (line 396, page 9)

Previous data indicates that the shortening of the eye axis of the human eye by 1 mm may bring about 2.3-2.5 diopters myopia correction [Ma, F. et al. *Clin. Experiment. Ophthalmol.* **42**, 190-197 (2014); Weihua, M. et al. *Ophthalmologica* **225**, 127–134 (2011)].

Table R2 | Classification of myopia, emmetropia, and hyperopia by spherical equivalent objective error and Axial Length.

	Myopia ³		Emmetropia ⁴	Hyperopia ⁵
The spherical equivalent objective error (Dioptres)	Low myopia	High myopia	-0.5 D to +0.5 D	$\geq +0.5$ D
	≤ -0.5 D to -5.0 D	≤ -5.0 D to -10.0 D		
Axial length (mm)	24 – 25.7	≥ 25.7	22 - 24	< 22

3. Morgan, I. G., Ohno-Matsui, K. & Saw, S. M. Myopia. *Lancet* **379**, 1739-1748 (2012).

4. Qian F., Hongxia W., Zhixin J., Axial length and its relationship to refractive error in Chinese university students. *Contact Lens Anterior Eye* **45**,101470 (2022).

5. Lourdes L., Sergio B., Daniel C., Carlos D., Susana M. Myopic versus hyperopic eyes: axial length, corneal shape and optical aberrations. *J. Vision* **4**, 5 (2004).

Comment 3. In Figure 4J, authors should explain the extent to which axial elongation occurs according to Young's modulus after cross-linking (CXL).

Reply 3. To clarify our terminology and avoid confusion with corneal collagen crosslinking (CXL), we have updated our manuscript to use the abbreviation SCXL for scleral crosslinking. Previous biomechanical research has consistently shown that scleral stress-strain parameters and Young's modulus in myopic eyes are notably lower than in normal eyes. This difference is primarily attributed to the lack of stable intramolecular and intermolecular bonds in the collagen of the sclera in myopic conditions. In SCXL, a photochemical reaction is induced in scleral collagen fibers using a photosensitizer, aimed at enhancing Young's modulus, thereby increasing the

scleral stress and overall biomechanical stability. Our study presents compelling evidence of the effectiveness of SCXL. At 7 days post-surgery, the Young's modulus in the SCXL group exhibited a substantial increase, recorded at 132.94% relative to the control group, as illustrated in Figure 6d. This marked improvement is a strong indicator of the initial success of the SCXL procedure in augmenting the stiffness of the scleral tissue. Furthermore, by 22 days post-surgery, Young's modulus showed an additional increase, reaching an impressive 387.717% compared to the control group, as shown in **Fig. R4**. This ongoing enhancement over time underscores the persistent and significant impact of the SCXL treatment in strengthening the scleral tissue. The increases in Young's modulus observed at both 7- and 22-days post-surgery highlight the efficacy of scleral collagen cross-linking (SCXL) in improving the biomechanical properties of the sclera. In this study, the specific formulas for quantifying strength enhancement and axial length modification are not provided, previous research indicates that the increased strength from treatment remains relatively stable up to 8 months, without a significant downward trend [Wollensak, G. et al. *Acta Ophthalmol.* **87**, 193-198 (2009)]. This stability is particularly relevant in the management of high myopia, where a stronger sclera plays a vital role in controlling axial elongation and slowing the progression of the disease.

Fig. R4 | Young's modulus of rabbit eyes in the control and SCXL groups. Statistical comparisons were performed using unpaired t test; *p=0.0434, **p=0.0059. n=4 eyes per group. Data are presented as means \pm SD.

Based on the Referee's comment, we have replaced **Fig. 6d** in the manuscript by **Fig. R4**. And we have incorporated revisions in the manuscript (line 411, page 10).

Comment 4. In Figure 5K, IOP increased by 6.1 mm/Hg. Authors should discuss the possible negative side effect such as glaucoma by the increase in IOP.

Reply 4. We express our gratitude to the Reviewer for their insightful observations and the valuable suggestion to monitor intraocular pressure (IOP) in our study. We concur that vigilant monitoring of IOP post-operatively is crucial for preventing potential glaucoma development due to elevated intraocular pressure. In response to this suggestion, we have augmented our data with continuous IOP monitoring over a

period of 22 days. During this extended monitoring period, we observed an initial increase in IOP in the six rabbits subjected to surgery, which we attribute to the surgical procedure itself and the subsequent implantation of the patch. This elevation in IOP was marked and reached a peak on the day following the surgery, as shown in **Fig. R5**. It's notable that the maximum post-surgery intraocular pressure was around 23.6 mmHg, significantly lower than the average intraocular pressure seen in glaucoma cases (**Table R1**).

Fig. R5 | Monitoring of intraocular pressure (IOP) changes over a 22-day period: (0 -) OD before surgery, (0+) OD after surgery (Patch retained group).

It is important to emphasize that, following the initial post-surgical elevation, the intraocular pressure (IOP) values began to normalize by the third day after the surgery. During the comprehensive 22-day monitoring period, we diligently recorded the IOP readings (n = 6). We observed a stabilization of IOP values close to preoperative levels starting from the third day post-surgery, with this stability maintained consistently up to the 20th day. This trend is a positive indication, suggesting that the initial surge in IOP observed post-surgery is a temporary response rather than a lasting complication. In terms of specific values, the highest recorded IOP was 23.6 mmHg, while the lowest was 8.2 mmHg. Generally, the IOP tended to fluctuate within a range of 8.3 to 12.6 mmHg. These detailed findings, including the range and stabilization of IOP values post-surgery, are presented in the revised manuscript (see Page 10, Line 435) and are further illustrated in Supplementary Figure 23. This data provides valuable insights into the transient nature of IOP changes following surgical intervention and patch implantation, reinforcing the potential safety and efficacy of this treatment method.

Table R3 | Intraocular pressure reference value.

IOP (mmHg)	This research	Glaucoma	Hypotony
------------	---------------	----------	----------

Control	9.0-13.8	30-37 ¹	$\leq 6^2$
Post-implantation (0+)	16.6-23.6		
22 days	8.3-12.6		

1. Ueno, A., Tawara, A., Kubota, T. et al. Histopathological changes in iridocorneal angle of inherited glaucoma in rabbits. *Graefe's Arch. Clin. Exp. Ophthalmol.* **237**, 654–660 (1999).

2. Adan V., Nikos S., Stéphane B., André M., Alexandre M., Sylvain R. In vivo testing of a novel adjustable glaucoma drainage device. *Invest. Ophthalmol. Vis. Sci.* **55**, 7520-7524 (2014).

Comment 5. How was the patch removed? If the patch remained, it might cause the negative side effect by the immune response. The patch size seems too large to be biocompatible.

Reply 5. We appreciate the reviewer inputs regarding the removability of the patch, prompting us to clarify the application of the patch in patients with high myopia. In cases where high myopia continues to progress, leading to excessive elongation of the eye axis without associated fundus lesions, the patch can be strategically utilized. Its role in these scenarios is to perform scleral collagen cross-linking, thereby strengthening the posterior pole sclera. This intervention is crucial in preventing the thinning and relaxation of scleral tissue caused by the elongated eye axis, thus averting the progression towards pathological myopia. For patients experiencing only progressive high myopia without pathological changes, the device is designed to be removable immediately after the completion of postoperative collagen cross-linking. This approach is suitable when the primary objective is to surgically reinforce the sclera, without the presence of pathological myopia.

High myopia carries a substantial risk of progression to pathological myopia if left uncontrolled. Pathological myopia can lead to severe ocular conditions, notably myopic maculopathy and myopic traction maculopathy. The development of a specialized patch, designed to cover the posterior pole of the eye, offers a proactive solution in managing these risks. This patch provides continuous support to the eye, specifically targeting the inward contraction of the relaxed posterior pole structure. It exerts a controlled force to maintain the structural integrity of the eye. The patch extends beyond basic vision correction, playing a pivotal role in posterior scleral reinforcement to mitigate complications associated with high and pathological myopia [Zhu, Z. et al. *Clin. Exp. Ophthalmol.* **37**, 660-663 (2009); Zhu, S. Q. et al. *Br. J. Ophthalmol.* **102**, 1701-1704 (2018); Zhu, S. Q. et al. *Br. J. Ophthalmol.* **100**, 1470-1475 (2016)]. It shows potential in reducing vitreous body traction on the macula. This reduction could significantly lower the risk of retinal detachment, a common and serious complication in advanced myopia. Furthermore, the patch may help in preventing choroidal atrophy,

a critical issue in pathological myopia that can lead to further vision deterioration. By addressing these structural issues at their source, the treatment aims to halt the progression of the disease, potentially leading to improved visual outcomes, as shown in **Fig. R6**.

Fig. R6 | Depiction of patch treatment in high myopia progression.

In addressing the management of fundus lesions resulting from high myopia, our approach with the patch is designed to provide continuous stress to the fundus. This sustained application of stress is crucial in preventing further relaxation of the posterior pole, thereby aiming to halt the progression of conditions such as choroidal atrophy or the exacerbation of vitreous traction maculopathy. The rationale behind this approach is based on the understanding that stabilizing the structural integrity of the fundus is key in managing the complications of high myopia. To illustrate the practicality and feasibility of our patch in clinical settings, we refer to data on the sizes of surgical instruments currently employed in various clinical treatments. These instruments range in size from 5 – 30 mm (**Fig. R7**) and have been proven to be stably retained behind the human eye [Susvar P et al. *Indian J. Ophthalmol.* **66**, 1772-1784 (2018)]. This evidence is significant as it suggests that a patch falling within this size range is likely to be successfully retained in the human body, thus providing the necessary long-term treatment. The accompanying figure, which details the size range of these surgical instruments, serves as a benchmark for designing our patch. By ensuring that the patch dimensions are within this established size range, we aim to optimize the likelihood of successful retention and functionality in the human eye. This approach not only aligns with current clinical practices but also leverages existing knowledge and experience in ocular treatments to enhance the safety and effectiveness of our novel treatment strategy.

TYPE OF MACULAR BUCKLE	SILASTIC SPONGE ROD 	ANDO'S PLOMBE 	L SHAPED MACULAR BUCKLE 	ADJUSTABLE MACULAR BUCKLE 	DEVIN MORINS T SHAPED BUCKLE 	AJL MACULAR BUCKLE BUCKLE DETAILS	12*2mm Silastic sponge	Silicone plate with stainless steel wire 5 different sizes (21-29mm length) Plate size 5*5mm	Silicone sponge 3cm long and 5mm thick and 7mm large with malleable Ti stent (15mm Silicone rubber)	radial exoplant. Handle (2*2*10mm) with plate quadrangular (4*4mm) or circular/meridional (5mm)	T Shaped (Morin Devin T- shaped macular wedge)	PMMA material covered by silicone The arm's length and curvature is customized.
PROCEDURE	 All 4 recti and IO tagged, SO severed Globe exposed to bring the posterior pole into view. Two mattress sutures placed at posterior pole, nasal one around fovea(5/6-0 Dacron)	 Lateral cantotomy done Check ligaments severed -Suture placed 16 mm from limbus in between SO and IO to hold plate -SRF drainage required -IO to visualise indent	 Buckle is slid head down along the lateral rectus muscle from the STQ. The tail is sutured with mattress suture (6-0 µ) -Chandler light used -0.3ml SF6 used in all	 SR, LR, IR tagged. LR disinserted. Suture (5-0 mersilene) passed through 2 lateral wing-lets. After positioning plate under macula with support from handle, the indentation is done and sutures finally tied anterior to equator nasal to SR, IR	 Tag the SR, IR, LR and IO with 2-0 Mersilene. 2mm band is passed under LR, IO and IR and fixed inferiorly. Plate is threaded and superior band passed under SR Plate passes under LR and macular indent finalised. Free end of plate sutured under LR and band nasal to SR	 All recti tagged and buckle is slid underneath the LR muscle(25G optical fibre fixed for illumination) or from ST quadrant. After positioning fixed with 5-0 ethibond under LR.
ADVANTAGE		Advantages: easy to insert the buckle, shape memory, customization of buckle with its embedded wire and eliminates the need for sutures on the staphylomatous sclera near macular area	It can be easily prepared in the operation theatre and the technique can be performed without the need of specially designed buckles, which are not available easily in all countries.	No direct posterior suture, buckle indent can be titrated by pulling two sutures.	Allows the adjustment for lengthening or shortening the band anteriorly while sliding the macular plate in the coronal plane; does not require posterior sutures or direct access to the posterior pole, no need for muscle disinsertion	Customized to individual patients eye, supplied with an optic fiber light probe for accurate positioning of the head under the fovea
DISADVANTAGE	Direct suturing at posterior pole, difficult globe exposure, Fat prolapse, vessel injury, globe perforation	Stiffness, limitations in adjustment of height, long-term safety of metal wire.	Include the unknown long-term safety of the titanium inside the orbit	Requires muscle disinsertion.	Possibility of improper alignment under the fovea.	Price and availability

Fig. R7 | Overview of different types of scleral reinforcement implants [Susvar P et al. *Indian J. Ophthalmol.* **66**, 1772-1784 (2018)].

Clinical trials will be essential to evaluate the treatment's safety and efficacy in human patients. The dimensions of our patch – 4 mm in thickness and 19 mm between the two legs – were specifically designed for ease of implantation in the scleral region at the rear pole of a rabbit's eye. Given the relatively larger size of human eyes compared to

rabbits', we anticipate that the application of the device in human eyes would be feasible and potentially safer. However, this assumption needs to be rigorously tested in clinical trials to ensure compatibility and safety for human patients.

Fig. R8 | Wireless eye modulation high myopia patch. (a and b) Flexible micro-fabricated circuit design. Scale bar, 3 mm. (c - e) The flexible and transparent PDMS encapsulates the system. Scale bar, 3 mm. (f) The whole system weight 0.41 g.

Our method aims to reduce these risks by minimizing the duration of surgical exposure. Traditional methods usually do not allow for postoperative adjustments, which can lead to challenges in precision. Wireless technology is intended to offer more control and accuracy in this aspect. The biocompatible, flexible materials (PDMS) can reduce the likelihood of adverse reactions and to ensure compatibility with the eye's natural structure. Postoperative analyses, including pathology and immunohistochemistry, have not indicated significant inflammation or complications. Additionally, intraocular pressure has generally remained stable. In conclusion, while our approach incorporates advancements in technology and materials, we recognize the importance of ongoing research and evaluation to fully ascertain its effectiveness and safety (see the **Supporting Fig. S22-29**). Our findings so far are promising, but we continue to approach our research with diligence and an understanding of the complexities involved in ocular treatments.

Following the Referee's comment, we have incorporated revisions in the manuscript (line 35, page 1; line 422, page 10; line 489, page 11), and more details can be found in **Supporting Fig. S22-29**.

Minor issue:

Comment 1. In Figure 4H, authors should provide the scale bar.

Reply 1. Thank you for pointing these out to us. We have corrected these about the scale bar in the revised manuscript (See Page 24, Line 908 in the revised manuscript,

Fig. 4h and Supplementary Fig. 16).

Comment 2. Authors should indicate the molecular weight of PVP used for microneedles in the section of Materials and Method.

Reply 2. Thank you for pointing these out to us. We have added the molecular weight of PVP in the revised manuscript (See Page 13, Line 576 in the revised manuscript).

Comment 3. Authors should describe the size and thickness of the patch.

Reply 3. The dimensions of our patch – 4 mm in thickness and 19 mm between the two legs – were specifically designed for ease of implantation in the scleral region at the rear pole of a rabbit's eye.

Based on the Referee's comment, we have replaced **Supplementary Fig. S2** in the manuscript by **Fig. R8**. We have incorporated revisions in the manuscript (line 146, page 4).

REVIEWER COMMENTS

Reviewer #1 (Remarks to the Author):

The authors have made great work to address my comments. The article can be published.

Reviewer #2 (Remarks to the Author):

This manuscript has been greatly improved after revision, but authors should address the following issues clearly.

[1] Although authors responded that the eye-modulation patch can be used for the treatment of myopia, the rationale and the advantages of this system are still unclear in comparison with conventional clinical systems and smart contact lens systems. Especially, this system might suffer from the patient incompliance because of its high invasiveness. In addition, this disposable system would not be used repeatedly for multiple drug delivery.

[2] Authors described that this system was powered by the ultrasound, but it is hard to find the control systems of soft actuators and micro LEDs. Authors should describe the control methods to operate soft actuators and micro LEDs, respectively.

[3] For the characterization of riboflavin delivery, authors should describe the total loading amount and the full delivery amount of riboflavin.

[4] Although the patch was fixed by the concavity structure on the posterior sclera, the patch would be easily moved by the eyeball, resulting in the optical nerve damage and the reduced therapeutic effect. Authors should describe the detailed fixation method of the patch.

[5] To further characterize the soft electrochemical actuator, authors should discuss the effect of loaded liquid amount on the actuation. This is very important, because it can affect the thickness of the patch.

[6] Authors should provide the penetration data of blue light to confirm the feasibility of light induced crosslinking of riboflavin. In addition, the arrangement and intensity of LEDs would affect the therapeutic effect and the crosslinking density.

POINT-BY-POINT REPLY TO PEER REVIEW COMMENTS

(Manuscript Number: NCOMMS-23-37475B)

REVIEWER #1 COMMENTS:

The authors have made great work to address my comments. The article can be published.

REPLY TO REVIEWER #1:

Authors: Thanks for your comment and recognition. Your recommendation for publication in *Nature Communications* is greatly appreciate.

REVIEWER #2 COMMENTS:

This manuscript has been greatly improved after revision, but authors should address the following issues clearly.

Authors: We acknowledge Reviewer #2 for their effort in assessing our manuscript. We would like to highlight that the Reviewer's comments helped enormously for the improvement of our manuscript. Below we addressed the comments point-by-point.

REPLY TO REVIEWER #2:

Comment 1. Although authors responded that the eye-modulation patch can be used for the treatment of myopia, the rationale and the advantages of this system are still unclear in comparison with conventional clinical systems and smart contact lens systems. Especially, this system might suffer from the patient incompliance because of its high invasiveness. In addition, this disposable system would not be used repeatedly for multiple drug delivery.

We thank the Referee for these valuable comments. We divide into the following two points to address the question:

Comment 1-1. Although authors responded that the eye-modulation patch can be used for the treatment of myopia, the rationale and the advantages of this system are still unclear in comparison with conventional clinical systems and smart contact lens systems. Especially, this system might suffer from the patient incompliance because of its high invasiveness.

Reply 1-1.

We are very grateful to the reviewer for raising this confusing question, and we will elaborate more on the clinical significance of the patch in this response. At present, there are many physical or minimally invasive methods to improve the refractive error or refractive error of myopia, such as contact lenses, excimer laser surgery or femtosecond laser myopia correction surgery. However, there are a large number of high myopia patients with progressive elongation of the eye axis and rapid increase of myopia in clinical work. **These patients will not stop the deterioration of the fundus due to whether they have refractive correction or not.** With the extension of the eye axis, the retinal choroid of the fundus, especially the macular retina, will show various pathological changes, the most common of which are macular atrophy and thinning, retinal splitting, macular hole and posterior scleral staphyloma. There are two main treatment methods for this kind of high myopia progression. **One is posterior scleral cross-linking**, which is a chemical collagen cross-linking method by using drugs and light to enhance the strength of the sclera and organize the further growth of the eye axis. This surgery is currently carried out in animal experiments, and it strengthens the strength of the posterior sclera itself. It does not provide a lasting external support force for the posterior sclera. For the posterior sclera that becomes abnormally loose due to the serious growth of the eye axis, it is not enough to simply increase the scleral strength by collagen cross-linking. At this time, **another** commonly used **posterior scleral reinforcement (Posterior scleral reinforcement, PSR)** is needed for treatment, which is to use different X-shaped or Y-shaped materials to "pad" the posterior sclera to achieve the purpose of strengthening the posterior sclera. The materials include dura mater, tendon, aorta and allogeneic sclera. According to the published research, clinical doctors most often use allogeneic sclera as the implant material. **It was mainly carried out in Europe and China at the earliest, mainly for adult and child high myopia patients. It has been used in clinical practice for the treatment of high myopia, macular lesions and other diseases for more than 20 years [1], and has been proven to be a safe and effective method for the treatment of high myopia macular lesions [2, 3].** Some studies have shown that after one year of PSR treatment, myopic macular splitting was relieved and the macula reattached. It can avoid further intraocular surgery and has the additional benefit of preventing the extension of the eye axis and the deepening of myopia. In a prospective controlled study lasting for 2 years, compared with the traditional vitrectomy, the success rate of posterior scleral reinforcement for macular hole associated macular detachment was higher [4]. It can provide a lasting support force to prevent the further growth of the eye axis. In recent years, many teams in the world have participated in the research in this field, and tried various top pressure materials (implants), such as expanded polytetrafluoroethylene, silicone, silicone sponge, titanium plate, wide neck membrane and allogeneic sclera, etc. [1,5,6]. Some studies have demonstrated that the best-corrected visual acuity (BCVA) of highly myopic eyes (experimental group) treated with PSR was significantly better than that of eyes wearing contact lenses (control group). Three years postoperatively, the axial length of the PSR-treated group was statistically different from that of the control group.

The results indicated that the scleral reinforcement could slow down the progression of myopia [7,8,9].

Based on the above research, our designed **eye-modulation patch integrates the advantages of scleral cross-linking and posterior scleral reinforcement surgery:**

1. It can enhance the hardness of the sclera and reduce the extension of the eye axis by scleral cross-linking. The concern raised regarding the limitations of existing scleral collagen crosslinking methods is indeed pertinent. Traditional techniques often face challenges in fully exposing the posterior pole sclera during surgery, which restricts most scleral crosslinking procedures to the equatorial region, precluding effective treatment of the posterior pole scleral region. Some studies have attempted to address this by inserting an optical fiber into the posterior pole for collagen crosslinking. However, these attempts often encounter inefficiencies due to inadequate adhesion between the optical fiber and the sclera, as well as obstruction of light propagation by blood flow, resulting in suboptimal collagen crosslinking efficiency. In contrast, the current study introduces a patch that is sutured and firmly attached to the sclera. This approach ensures smooth light transmission to the treatment area during the drug-induced collagen crosslinking process, facilitating effective scleral collagen crosslinking.

2. The patch can provide therapeutic effects for posterior scleral reinforcement surgery. It can not only adjust the eye axial length by external ultrasound, but also provide persistent support force for the posterior sclera on the basis of scleral collagen cross-linking, preventing excessive relaxation and expansion of the sclera. For patients with high myopia whose eye axis is still growing, the patch can be used for scleral collagen cross-linking treatment (Patch removal). This can increase the strength of the patient's posterior sclera, thereby preventing the further elongation of the eye axis and the occurrence of pathological lesions. For patients who have developed pathological myopia, the patch can shorten the eye axial length and provide long-term support force for the posterior part of the eye after treatment (Patch retained), playing the role of posterior scleral reinforcement, and preventing the condition from further deteriorating.

We value the reviewer's concern regarding patient compliance. It is important to recognize that not all patients with high myopia require surgical intervention. The necessity for surgery primarily arises in cases of progressive high myopia and in patients exhibiting fundus lesions associated with high myopia. In the treatment of progressive high myopia and pathological myopia, invasive surgical interventions frequently stand as the sole viable option to halt the progression of these conditions. Despite the invasive nature of these surgeries, their potential to prevent further worsening of the patients' visual impairments often makes them a necessary course of action in the management of early and advanced high myopic conditions. In such scenarios, the potential benefits of surgical intervention often significantly outweigh the risks associated with invasive procedures.

This study is still in its early stages. The functionality of our device in terms of material measurement and drug delivery also requires further verification and improvement in upcoming human experiments. We are committed to advancing this research to provide more comprehensive treatment options for myopia.

Following the Referee's comment, we have incorporated revisions in the manuscript (line 35, page 1).

Comment 1-2. In addition, this disposable system would not be used repeatedly for multiple drug delivery.

Reply 1-2. The process of scleral crosslinking involves riboflavin penetrating the tissue and subsequently forming new bonds between scleral collagen fibers upon exposure to blue light. This results in increased strength and a more compact arrangement of the collagen fibers, creating a denser structure. Importantly, current research indicates that this crosslinking process is irreversible, and there is almost no necessity for secondary or multiple administrations of the drug post-surgery. For example, in corneal collagen cross-linking experiments that operate on a similar principle, studies have indicated that the increased scleral strength post-procedure maintains its therapeutic efficacy for up to about five years [10]. However, it is imperative to note that more extensive samples and research are required to firmly establish the effectiveness of scleral collagen cross-linking therapy especially in the future human experiments.

Comment 2. Authors described that this system was powered by the ultrasound, but it is hard to find the control systems of soft actuators and micro LEDs. Authors should describe the control methods to operate soft actuators and micro LEDs, respectively.

Reply 2. We acknowledge the necessity for enhanced clarity regarding our initial explanation of the circuit control mechanism in our device and appreciate the opportunity for further elaboration. The operation of each individual circuit in our system hinges on meticulous control via an external ultrasonic source, as comprehensively depicted in Supplementary Fig. 3. Upon activation of the ultrasonic source, it propagates an ultrasonic signal. This signal, when directed towards the targeted receiver piezoelectric transducer (PZT) of a specific circuit, is received by the PZT, thereby triggering the activation of that particular circuit. This process enables the circuit to fulfill its designated role. Such a mechanism ensures a high level of precision and efficiency in the functioning of our system, contributing to its overall efficacy and reliability.

In our study, the design of the PZT spacing is carefully tailored to align with the dimensions of the rabbit eye, ensuring optimal device performance. As demonstrated in experiment and corroborated by simulations (refer to **Fig. R1**), this design is instrumental in preventing unintended activation of the device. We observed that ultrasonic propagation exhibits minimal attenuation in the central direction perpendicular to the probe plane, while significant loss occurs in transverse propagation. The effective operational diameter around the probe center is approximately 10 mm. We have deliberately established a minimum separation of over 15 mm between adjacent PZT. This precautionary measure is designed to surpass the average diameter

of a rabbit's eyeball, effectively mitigating the risk of unintentional triggering of neighboring functions.

Fig. R1 | Simulation of acoustic pressure distribution in the eyeball.

Following the Referee's comment, we have incorporated revisions in the manuscript (line 153, page 4 and line 210, page 5).

Comment 3. For the characterization of riboflavin delivery, authors should describe the total loading amount and the full delivery amount of riboflavin.

Reply 3. We appreciate the reviewer's insightful feedback, which has been instrumental in sharpening the focus of our study on riboflavin delivery via microneedles. The referenced prediction model results indicate a general increase in crosslinking with rising light intensity and photosensitizer drugs concentration, spanning ranges of 0 - 424 mW cm⁻² and 0 - 100 mM, respectively [11]. It is important to note that our study utilizes solid-state drug needles, whereas existing research predominantly employs liquid solutions for administration. Bearing this in mind, we have referred to a comparable range and, based on experimental data, tested three distinct concentrations: 1%, 5%, and 10%. After careful consideration and evaluation of the results, we selected the 10% concentration for use in our experiments. In the current research, a single dose of approximately 4.8 mg (10%) riboflavin was administered. This specific dosage was determined based on the assessment of fluorescence intensity, H&E tissue staining, and Young's modulus measurements across multiple riboflavin concentrations (1%, 5% and 10%), as detailed in **Fig. R2**. It was observed that lower concentrations of riboflavin resulted in diminished collagen crosslinking effects and reduced fluorescence intensity.

After a diffusion period of 20 min, a portion of the riboflavin successfully infiltrated the scleral tissue, facilitating the collagen cross-linking process. Due to the limited research on the quantification of drug delivery in scleral crosslinking, precise values for the exact amount of riboflavin delivered to the tissues are not established at this stage. However, previous studies suggest that fluorescence intensity is a reliable measure of drug delivery, exhibiting a direct correlation with the quantity of drug delivered. In this study, the post-drug release fluorescence intensity was found to increase by 2.691 times relative to the levels before release, as demonstrated in **Fig. R3**. This significant enhancement in fluorescence intensity (The distance from the administration side is ~200-400 μm) is comparable to that seen in established corneal collagen crosslinking procedures, indicating effective drug delivery [12]. Furthermore, our observations 7 days post-operation, after the removal of the patch (**Fig. R4**),

indicated that while a very small amount of riboflavin remained in the retractor bulbi muscle at the treatment site, the majority of the riboflavin was successfully delivered.

It's crucial to note that research into drug delivery and dosing for scleral collagen crosslinking is still in its nascent stages. Consequently, the optimal drug dosage for this application remains an area for further exploration, which may vary depending on the animal model used. Our findings contribute to this ongoing research, providing a foundation for future studies to refine drug delivery methods and dosages for scleral collagen crosslinking.

Fig. R2| **a** Hematoxylin and Eosin (H&E) staining (Scale bar, 50 μm), **b** fluorescence intensity measurement (Scale bar, 100 μm) and **c** Young's modulus assessments of porcine tissue with riboflavin microneedles of various concentrations (Scale Bar: 50 μm)

Fig. R3| Fluorescence intensity of drug release from microneedles into sclera over time (0-30 min). n=5 per group.

Fig. R4 | After fixing rabbit eyes with eyeball fixation fluid, the patch was removed, revealing that the drug was almost entirely delivered, with just a minimal amount penetrating the retractor bulbi muscle. Scale bar, 2 mm

Following the Referee's comment, we have incorporated revisions in the manuscript (line 597, page 14).

Comment 4. Although the patch was fixed by the concavity structure on the posterior sclera, the patch would be easily moved by the eyeball, resulting in the optical nerve damage and the reduced therapeutic effect. Authors should describe the detailed fixation method of the patch.

Reply 4. Upon identifying the appropriate experimental site, the patch was carefully positioned at the designated location and pressed against the exposed sclera. Once accurately placed, the three legs of the patch were securely sutured to the sclera using a 5-0 Coated Vicryl Plus synthetic absorbable suture (Ethicon, Inc. USA), as shown in **Fig. R5**. This procedure ensured the patch's firm attachment to the posterior polar scleral region, thus maintaining its correct position. The stability provided by the secure suturing of all three legs to the scleral tissue ensures that the patch remains in place without any shift or deviation, even during movements of the eyeball. In parallel, devices based on a similar principle to the scleral top pressure device have been clinically implemented. Postoperative MRI evaluations have confirmed the stability of these devices, as they remain securely fixed at their sutured positions [13].

Fig. R5 | Suturing of the patch's three legs to the sclera using a 5-0 suture. Scale bar, 3 mm.

Following the Referee's comment, we have incorporated revisions in the manuscript (line 661, page 15). And we have added **Supplementary Fig. 34** in the Supplementary

file by **Fig. R5**.

Comment 5. To further characterize the soft electrochemical actuator, authors should discuss the effect of loaded liquid amount on the actuation. This is very important, because it can affect the thickness of the patch.

Reply 5. We appreciate the reviewer's insightful question and are thankful for the opportunity to clarify and expand upon our analysis and discussion of liquid load. This aspect is indeed crucial for the thickness of the patch.

In our design, the patch's area is 14 mm² to cater to the specific needs of the treatment site. Considering the electrolysis duration of 6 min, a sufficient volume of electrolyte solution is essential for uninterrupted operation. In scenarios with limited electrolyte solution volume, the process may prematurely cease due to inadequate liquid to separate the bubbles from the electrodes. This insufficiency could lead to bubbles enveloping the electrode surface, thereby halting the electrolytic reaction. As demonstrated in **Fig. R6**, with an electrolyte solution volume of 15 μ l, the liquid chamber's thickness measures 1.07 mm, but the reaction ceases after 270 s as bubbles cover the electrode surface. Conversely, increasing the electrolyte solution volume to 25 μ l results in a liquid chamber height of 1.78 mm, allowing the electrolytic reaction to proceed smoothly for up to 420 s without interruption.

Our experiments indicate that the minimum volume capacity of electrolyte solution in the liquid chamber is 25 μ l. To ensure the reliability and stability of the experiment, we designed the chamber to hold a volume of 35 μ l, corresponding to a chamber height of 2.5 mm and an overall patch thickness of approximately 4 mm. This configuration allows for further optimization based on experimental needs to achieve optimal functionality.

Fig. R6 | Minimum liquid capacity in the chamber – 25 μ L limit.

Following the Referee's comment, we have incorporated revisions in the manuscript (line 233, page 6). And we have added **Supplementary Fig. 9b** in the Supplementary file by **Fig. R6**.

Comment 6. Authors should provide the penetration data of blue light to confirm the feasibility of light induced crosslinking of riboflavin. In addition, the arrangement and

intensity of LEDs would affect the therapeutic effect and the crosslinking density.

We thank the Referee for these valuable comments. We divide into the following two points to address the question:

Comment 6-1. Authors should provide the penetration data of blue light to confirm the feasibility of light induced crosslinking of riboflavin.

Reply 6-1. We appreciate the valuable insights provided. The rabbit sclera, being roughly half as thick as pig sclera and containing less melanin (**Fig. R7b**), exhibits slightly stronger light transmission than pig sclera (**Fig. R7c**). Photographic evidence and transmittance measurements under various light intensities demonstrated that, at approximately 30 mW cm^{-2} light intensity before penetration, the transmission rate through sclera and choroidal tissues was 55.46% in rabbits and 15.40% in pigs (**Fig. R7d** and **e**). Given the structural similarity between pig and human eyes, these results indicate that only low-intensity light reaches the fundus post-penetration, thus confirming the fundus' safety during the experiment. Furthermore, a decrease in fluorescence intensity with deeper riboflavin penetration (**Fig. R8**) suggests a reduction in concentration. This implies that effective collagen crosslinking mainly occurs in about one-third of the outer sclera [11]. This finding confirms that achieving collagen cross-linking does not necessitate full, high-intensity light penetration through the sclera. Instead, enhancing the mechanical strength of the lateral sclera tissue is sufficient to achieve the desired therapeutic effect.

Fig. R7 | Light transmittance characteristics. **a** Intensity of blue μ -LED at varying current output. **b** Optical images of sclera of rabbit and porcine. Abundant choroidal melanin in porcine sclera. **c** Comparative diagram of light penetration in rabbit and porcine sclera: highlighting the higher pigment content in porcine sclera's choroid, greater light absorption, and reduced light transmission to the fundus compared to rabbit

sclera. Variations in blue light transmittance through the (d) rabbit and (e) porcine sclera at different intensities.

Fig. R8 | Fluorescent intensity image of porcine sclera with drug penetration. Scale bar, 100 μm .

Comment 6-2. In addition, the arrangement and intensity of LEDs would affect the therapeutic effect and the crosslinking density.

Reply 6-2. The reviewer's suggestions have been invaluable, leading to the inclusion of COMSOL simulations to demonstrate the impact of varying arrangements and distributions of $\mu\text{-LEDs}$. Due to the patch's structural design, the placement of $\mu\text{-LEDs}$ in the central area, which contains the electrolyte solution, is not feasible. Consequently, $\mu\text{-LEDs}$ are positioned around this area. Simulations indicate that employing only one or two $\mu\text{-LEDs}$ fails to provide adequate irradiation area and intensity necessary for effective collagen cross-linking, owing to limitations in lighting range and location. As evidenced in **Fig. R9**, such configurations result in uneven and insufficient light intensity distribution across the scleral layer, hindering the achievement of collagen cross-linking in the targeted region. Conversely, the arrangement of three $\mu\text{-LEDs}$ around the patch ensures comprehensive coverage and adequate cross-linking light intensity, reaching the middle layer of the sclera. This arrangement enhances the collagen cross-linking effect. **Fig. R10** illustrates the cross-linking outcomes at different light intensities, showing a correlation between increased light intensity and Young's modulus strength. Research supports the safety and reliability of using a light intensity not exceeding 20-30 mW cm^{-2} for collagen cross-linking [14]. This study leverages these insights to optimize the $\mu\text{-LED}$ arrangement on the patch, ensuring both the efficacy and safety of the collagen cross-linking process.

Fig. R9 | Simulation analysis: μ -LED arrangement and intensity distribution impact on porcine scleral tissue.

Fig. R10 | The relationship between blue light intensity and Young's modulus. The control (CTRL) group represents the condition with no light and no drugs, while the riboflavin (RF) group represents the condition with no light but with drugs. Statistical comparisons were assessed by one-way analysis of variance (ANOVA); **** $p < 0.0001$. $n = 3$ per group. Data are presented as means \pm SD.

Following the Referee's comment, we have incorporated revisions in the manuscript (line 354, page 8). And we have added **Supplementary Fig. 17** in the Supplementary file by **Fig. R7** and **R9**.

References

1. Sasoh, M. et al. Macular buckling for retinal detachment due to macular hole in highly myopic eyes with posterior staphyloma. *Retin.-J. Retin. Vit. Dis.* **20**, 445-449 (2000).
2. Zhao, X. J. et al. Macular buckling versus vitrectomy on macular hole associated macular detachment in eyes with high myopia: a randomised trial. *Br. J. Ophthalmol.* **106**, 582-586 (2022).
3. Zhao, X. J. et al. Three-year outcomes of macular buckling for macular holes and foveoschisis in highly myopic eyes. *Acta Ophthalmol.* **98**, E470-E478 (2020).
4. He, Q. et al. Posterior scleral reinforcement for the treatment of myopic traction maculopathy. *BMC Ophthalmol.* **22**, 273 (2022).
5. Wu, P. C. et al. Gore-tex vascular graft for macular buckling in high myopia eyes. *Retin.-J. Retin. Vit. Dis.* **37**, 1263-1269 (2017).
6. Susvar, P. & Sood, G. Current concepts of macular buckle in myopic traction maculopathy. *Indian J. Ophthalmol.* **66**, 1772-1784 (2018).
7. Shen, Z. M., Zhang, Z. Y., Zhang, L. Y., Li, Z. G. & Chu, R. Y. Posterior scleral reinforcement combined with patching therapy for pre-school children with unilateral high myopia. *Graefes Arch. Clin. Exp. Ophthalmol.* **253**, 1391-1395 (2015).
8. Chen, M. J., Dai, J. H., Chu, R. Y. & Qian, Y. F. The efficacy and safety of modified Snyder-Thompson posterior scleral reinforcement in extensive high myopia of Chinese children. *Graefes Arch. Clin. Exp. Ophthalmol.* **251**, 2633-2638 (2013).
9. Huang, W. L., Duan, A. L. & Qi, Y. Posterior scleral reinforcement to prevent progression of high myopia. *Asia-Pac. J. Ophthalmol.* **8**, 366-370 (2019).

10. Zhang, X. et al. A review of collagen cross-linking in cornea and sclera. *J. Ophthalmol.* **2015**, 289467 (2015).
11. Gerberich, B. G. et al. Computational modeling of corneal and scleral collagen photocrosslinking. *J. Control. Release* **347**, 314-329 (2022).
12. Ricardo L. et al. Ultrasound-enhanced penetration of topical riboflavin into the corneal stroma. *Invest. Ophthalmol. Vis. Sci.* **54**, 5908-5912 (2013).
13. Liu, B. et al. Macular buckling using a three-armed silicone capsule for foveoschisis associated with high myopia. *Retin.-J. Retin. Vitr. Dis.* **36**, 1919-1926 (2016).
14. Li, Y. et al. Ocular safety evaluation of blue light scleral cross-linking in vivo in rhesus macaques. *Graefes Arch. Clin. Exp. Ophthalmol.* **257**, 1435-1442 (2019).